# A unified framework for bandit multiple testing

**Ziyu Xu**
Department of Statistics and Data Science
Carnegie Mellon University, USA
xzy@cmu.edu

**Ruodu Wang**
Department of Statistics and Actuarial Science
University of Waterloo, Canada
wang@uwaterloo.ca

**Aaditya Ramdas**
Department of Statistics and Data Science
Machine Learning Department
Carnegie Mellon University, USA
aramdas@cmu.edu

## Abstract

In bandit multiple hypothesis testing, each arm corresponds to a different null hypothesis that we wish to test, and the goal is to design adaptive algorithms that correctly identify large set of interesting arms (true discoveries), while only mistakenly identifying a few uninteresting ones (false discoveries). One common metric in non-bandit multiple testing is the false discovery rate (FDR). We propose a unified, modular framework for bandit FDR control that emphasizes the decoupling of exploration and summarization of evidence. We utilize the powerful martingale-based concept of "e-processes" to ensure FDR control for arbitrary composite nulls, exploration rules and stopping times in generic problem settings. In particular, valid FDR control holds even if the reward distributions of the arms could be dependent, multiple arms may be queried simultaneously, and multiple (cooperating or competing) agents may be querying arms, covering combinatorial semi-bandit type settings as well. Prior work has considered in great detail the setting where each arm's reward distribution is independent and sub-Gaussian, and a single arm is queried at each step. Our framework recovers matching sample complexity guarantees in this special case, and performs comparably or better in practice. For other settings, sample complexities will depend on the finer details of the problem (composite nulls being tested, exploration algorithm, data dependence structure, stopping rule) and we do not explore these; our contribution is to show that the FDR guarantee is clean and entirely agnostic to these details.

## 1 Introduction to bandit multiple hypothesis testing

Scientific experimentation is often a sequential process. To test a single null hypothesis — with "null" capturing the setting of no scientific interest, and the alternative being scientifically interesting — scientists typically collect an increasing amount of experimental data in order to gather sufficient evidence such that they can potentially reject the null hypothesis (i.e. make a scientific discovery) with a high degree of statistical confidence. As long as the collected evidence remains thin, they do not reject the null hypothesis and do not proclaim a discovery. Since executing each additional unit of data (stemming from an experiment or trial) has an associated cost (in the form of time, money, resources), the scientist would like to stop as soon as possible. This becomes increasingly prevalent when the scientist is testing multiple hypotheses at the same time, and investing resources into testing one means divesting it from another.

For example, consider the case of a scientist at a pharmaceutical company who wants to discover which of several drug candidates under consideration are truly effective (i.e. testing a hypothesis of whether each candidate has greater than baseline effect) through an adaptive sequential assignment of drug candidates to participants. Performing follow up studies on each discovery is expensive, so the scientist does not want to make many "false discoveries" i.e. drugs that did not have an actual effect, but were proclaimed to have one by the scientist. To achieve these goals, one could imagine

35th Conference on Neural Information Processing Systems (NeurIPS 2021).

the scientist collecting more data for candidates whose efficacy is unclear but appear promising (e.g. drugs with nontrivial but inconclusive evidence), and stop sampling candidates that have relatively clear results already (e.g. drugs that have a clear and large effect, or seemingly no effect).

**Past work.** This problem combines the challenges of multiple hypothesis testing with multi-arm bandits (MABs). In a "doubly-sequential" version of the problem studied by Yang et al. [46], one encounters a sequence of MAB problems over time. Each MAB was used to test a single special placebo arm against several treatment arms, and if at least one treatment dominated the placebo, then they aimed to return the best treatment. Thus each MAB was itself a single adaptive sequential hypothesis test, and the authors aimed not to make too many false discoveries over the sequence of MAB instances.

This paper instead considers the formulation of Jamieson and Jain [19], henceforth called JJ, but our techniques apply equally well to the above setup. *To avoid confusions, note that our setup is very different from the active classification work of the same authors [17].* To recap, JJ consider a single MAB instance without a placebo arm (or rather, leaving it implicit), and try to identify as many treatments that work better than chance as possible, without too many false identifications. To clarify, we associate each arm with one (potentially composite) null hypothesis — for example, the hypothesis that corresponding drug has no (significant) effect. A single observed reward when pulling an arm corresponds to a statistic that summarizes the results of one experiment with the corresponding drug, and the average reward across many experiments could correspond to an estimate of the average treatment effect, which would be (at most) zero for null arms and positive for non-nulls. Thus, a strategy for quickly finding the arms with positive means corresponds to a strategy for allocating trial patients to drug candidates that allows the scientists to rapidly find the effective drugs.

However, the above corresponds to only the simplest problem setting. In more complex settings, it may be possible to pull multiple arms in each round, and observe correlated rewards. Further, the arms may have some combinatorial structure that allows only certain subsets of arms to be pulled. There could be multiple agents (eg: hospitals) pulling the same set of arms and seeing independent rewards (eg: different patients) or dependent rewards (eg: patient overlap or interference). Further, if some set of experiments by one scientist yielded suggestive but inconclusive evidence, another may want to follow up, but not start from scratch, instead picking up from where the first left off. Last, the MAB may be stopped for a variety of reasons that may or may not be in the control of the scientist (eg: a faster usage of funding than expected, or additional funding is secured). We dive in the details of these scenarios in Appendix D.3.

**Our contribution.** We introduce a modular meta-algorithm for bandit multiple testing with provable FDR control that utilizes "e-values" — or, more appropriately, their sequential analog, "e-processes" — a recently introduced alternative to p-values (or p-processes) by Ramdas et al. [32] for various testing problems, that are inherently related to martingales, gambling and betting [33, 14, 15, 44]. This work is the first to carefully study e-processes in general MAB settings, building on prior work that studied a special case [44]. We also are the first to extend the bandit multiple testing problem to the combinatorial bandit setting — JJ had previously only analyzed the problem in the single-arm, independent reward setting. Utilizing e-processes provide our meta-algorithm with several benefits. (a) For composite nulls, it is typically easier to construct e-processes than p-processes; the same holds when data from a single source is dependent. When combining evidence from disparate (independent or dependent) sources, it is also more straightforward to combine e-values than p-values (see Appendix D.3). (b) The same multiple testing step applies in all bandit multiple testing problems, regardless of all the various details of the problem setup mentioned in the previous paragraph. Consequently, FDR control in our meta-algorithm is agnostic to much of problem setup and can be proved in a vast array of settings. This is not true when working for p-values. In particular, the techniques for proving FDR control in JJ are highly reliant on the specific bandit setup in their paper. (c) The exploration step can be — but does not have to be — decoupled from the multiple testing (combining evidence) step. This results in a modular procedure that can be easily ported to new problem settings to yield transparent guarantees on FDR control.

By virtue of being a meta-algorithm, we do not (and cannot) provide "generic" sample complexity guarantees: these will depend on all of the finer problem details mentioned above, on the exploration algorithm employed, on which e-processes are constructed. Our emphasis is on the flexibility with which FDR control can be guaranteed in a vast variety of problem setups. Further research can pick up one problem at a time and design sensible exploration strategies and stopping rules, developing sampling complexity bounds for each, and these bounds will be inherited by the meta-algorithm.

However, we do formulate some generic exploration algorithms in Appendix C based on best arm identification algorithms [1, 22, 11, 18, 23, 10, 20].

When instantiated to the particular problem setup studied by JJ (independent, sub-Gaussian rewards, one arm in each round, etc.), we get a slightly different algorithm from them — the exploration strategy can be inherited to stay the same, but the multiple testing part differs. JJ use p-processes for each arm to determine whether that arm should be added to the rejection set, and correct for testing multiple hypotheses by using the BH procedure [6] to ensure that the false discovery rate (FDR), i.e. the proportion of rejections that are false discoveries in expectation, is controlled at some fixed level $\delta$. Adaptive sampling induces a peculiar form of dependence amongst the p-values, for which the BH procedure provides error control at an inflated level; in other words, one has to use BH at a more stringent level of approximately $\delta/\log(16/\delta)$ to ensure that the FDR is less than $\delta$. On the other hand, we use the e-BH procedure [44], an analogous procedure for e-values, which can ensure the FDR is less than $\delta$ without any inflation, regardless of the dependence structure between the e-values of each arm. Our algorithm has improved sample efficiency in simulations and the same sample complexity in theory.

**Formal problem setup.** We define the bandit as having $k$ arms, and $\nu_i$ as the (unknown) reward distribution for arm $i \in [k] = \{1,...,k\}$. Every arm $i$ is associated with a null hypothesis, which is represented by a known, prespecified set of distributions $\mathcal{P}_i$. If $|\mathcal{P}_i| = 1$, it is a 'point null hypothesis', and otherwise it is a 'composite null hypothesis'. Examples of the latter include "all $[0,1]$-bounded distributions with mean $\leq 0.5$" or "all 1-sub-Gaussian distributions with mean $\leq 0$" or "all distributions that are symmetric around 0" or "all distributions with median $\leq 0$". While we assume by default that all rewards from an arm are i.i.d., we also formulate tests for hypotheses on reward distributions that may violate this assumption in Appendix G. If $\nu_i \in \mathcal{P}_i$, then we say that the $i$-th null hypothesis is true and we call $i$ a null arm; else, we say $i$-th null hypothesis is false and we call it a non-null arm. Thus, the set of arms are partitioned into two disjoint sets: nulls $\mathcal{H}_0 \subseteq [k]$ and non-nulls $\mathcal{H}_1 := [k] \setminus \mathcal{H}_0$.

Let $\mathcal{K} \subseteq 2^{[k]}$ denote the subsets of arms that can be jointly queried in each round. At each time $t$, the algorithm chooses a subset of arms $\mathcal{I}_t \in \mathcal{K}$ to sample jointly from. The special choice of $\mathcal{K} = \{\{1\}, \{2\}, ..., \{k\}\}$ recovers the standard bandit setup, but otherwise this setting is known as combinatorial bandits with semi-bandit feedback [12]. We also consider the special case of full-bandit feedback (the algorithm sees all rewards at each time step) in Appendix D.1. We denote the reward sampled at time $t$ from arm $i \in \mathcal{I}_t$ as $X_{i,t}$. Let $T_i(t)$ denote the number of times arm $i$ has been sampled by time $t$, and $t_i(j)$ be the time of the $j$th sample from arm $i$.

We now define a canonical "filtration" for our bandit problem. A filtration $(\mathcal{F}_t)_{t \geq 0}$ is a series of nested sigma-algebras that encapsulates what information is known at time $t$. (We drop the subscript and just write $(\mathcal{F}_t)$ for brevity, and drop the parentheses when just referring to a single sigma-algebra at time $t$.) Define the *canonical filtration* as follows for $t \in \mathbb{N}$: $\mathcal{F}_t := \sigma(U \cup \{(i,s,X_{i,j}) : s \leq t, i \in \mathcal{I}_s\})$ and we let $\mathcal{F}_0 := \sigma(U)$ where $U$ is uniformly distributed on $[0,1]$ and its bits capture all private randomness used by the bandit algorithm that are independent of all observed rewards. Let $(\lambda_t)$ be a sequence of random variables indexed by $t \in \mathbb{N}$. $(\lambda_t)$ is said to be *predictable* w.r.t. $(\mathcal{F}_t)$ if $\lambda_t$ is measurable w.r.t. $\mathcal{F}_{t-1}$ i.e. $\lambda_t$ is fully specified given the information in $\mathcal{F}_{t-1}$. An $\mathbb{N}$-valued random variable $\tau$ is a stopping time (or stopping rule) w.r.t. to $(\mathcal{F}_t)$ if $\{\tau = t\} \in \mathcal{F}_t$ — in other words, at each time $t$, we know whether or not to stop collecting data. Let $\mathcal{T}$ denote the set of all possible stopping times/rules w.r.t. $(\mathcal{F}_t)$, potentially infinite. Technically, the algorithm must not just specify a strategy to select $\mathcal{I}_t$, but also specify when sampling will stop. This is denoted by the stopping rule or stopping time $\tau^* \in \mathcal{T}$.

Once the algorithm halts at some time $\tau$, it produces a rejection set $\mathcal{S}_\tau \subseteq [k]$. We consider two metrics w.r.t. $\mathcal{S}$: the FDR as discussed prior, and true positive rate (TPR), which is the proportion of non-nulls that are discovered in expectation. These two metrics are defined as follows:

$$\text{FDR}(\mathcal{S}_\tau) := \mathbb{E}\left[\frac{|\mathcal{H}_0 \cap \mathcal{S}_\tau|}{|\mathcal{S}_\tau| \vee 1}\right], \qquad \text{TPR}(\mathcal{S}_\tau) := \mathbb{E}\left[\frac{|\mathcal{H}_1 \cap \mathcal{S}_\tau|}{|\mathcal{H}_1|}\right].$$

We consider algorithms that always satisfy $\text{FDR}(\mathcal{S}_\tau) \leq \delta$ for any number and configuration of nulls $\mathcal{H}_0$ and any choice of null and non-null distributions. In fact, our algorithm will produce a sequence of candidate rejection sets $(\mathcal{S}_t)$ that satisfies $\sup_{\tau \in \mathcal{T}} \text{FDR}(\mathcal{S}_\tau) \leq \delta$. This is a much stronger guarantee than the typical setting considered in the multiple testing literature. On the other hand, TPR is a measurement of the power of the algorithm i.e. how many of the non-null hypotheses does the algorithm discover. Our implicit goal in the multiple testing problem is to maximize the number of true discoveries while not making too many mistakes i.e. keep the FDR controlled.

In hypothesis testing, the set of null distributions $\mathcal{P}_i$ for each arm $i$ is known, because the user defines the null hypothesis they are interested in testing. *When the null hypothesis is false, the non-null distribution can be arbitrary.* Consequently, we can prove results about FDR, but we cannot prove guarantees about TPR without several further assumptions on the non-null distributions, dependence across arms, etc. For a particular setting where we make such a set of assumptions, we demonstrate in Section 4 that we can prove TPR guarantees for algorithms within our framework. Hence, our FDR controlling framework is not vacuous as it includes powerful algorithms i.e. algorithms which make many true discoveries. However, our focus is primarily to show that the FDR control of our framework is robust to a wide range of conditions.

Finally, note that in bandit multiple testing, one does not care about regret. The problem is more akin to *pure exploration*, where we aim to find a $\mathcal{S}$ with $\mathrm{FDR}(\mathcal{S}_{\tau^*}) \leq \delta$ and large TPR as quickly as possible.

Now that we have specified the problem we are interested in, we can introduce our main technical tools for ensuring FDR control at stopping times: e-processes and p-processes.

## 2 Technical preliminaries

### 2.1 E-processes versus p-processes

An e-variable, $E$, is a nonnegative random variable where $\mathbb{E}[E] \leq 1$ when the null hypothesis is true. In contrast, the more commonly used p-variable, $P$, is defined to have support on $(0,1)$ and satisfy $\mathbb{P}(P \leq \alpha) \leq \alpha$ for all $\alpha \in (0,1)$ when the null hypothesis is true. To clearly delineate when we are discussing solely the properties of a random variable, we also use the terms "e-value" $e$ and "p-value" $p$ to refer to the realized values of a e-variable $E$ and a p-variable $P$ (their instantiations on a particular set of data). E-variables and p-variables are connected through Markov's inequality, which implies that $1/E$ is a p-variable (but $1/P$ is not in general an e-variable). Rejecting a null hypothesis is usually based on observing a small p-value or a large e-value. For example, to control the false positive rate at 0.05 for a single hypothesis test, we reject the null when $p \leq 0.05$ or when $e \geq 20$.

Since bandit algorithms operate over time, we define sequential versions of p-variables and e-variables. A p-process, denoted $(P_t)_{t \geq 1}$, is a sequence of random variables such that $\sup_{\tau \in \mathcal{T}} \mathbb{P}(P_\tau \leq \alpha) \leq \alpha$ for any $\alpha \in (0,1)$. In contrast, an e-process $(E_t)_{t \geq 1}$ must satisfy $\sup_{\tau \in \mathcal{T}} \mathbb{E}[E_\tau] \leq 1$ (let $E_\infty := \limsup_{t \in \mathbb{N}} E_t$ and $P_\infty := \liminf_{t \in \mathbb{N}} P_t$). These sequentially valid forms of p-variables and e-variables are crucial since we allow the bandit algorithm to stop and output a rejection set in a data-dependent manner. Thus, we must ensure the respective properties of p-variables and e-variables hold over all stopping times.

These concepts are intimately tied to sequential testing and sequential estimation using confidence sequences [31], but most importantly, nonnegative (super)martingales play a central role in the construction of efficient e-processes. To summarize, (a) for point nulls, all admissible e-processes are simply nonnegative martingales, and the safety property follows from the optional stopping theorem, (b) for composite nulls, admissible e-processes are either nonnegative martingales, or nonnegative supermartingales, or the infimum (over the distributions in the null) of nonnegative martingales. Associated connections to betting [45] are also important for the development of sample efficient algorithms and we discuss how we use betting ideas in Appendix F. We also discuss some useful equivalence properties of p-processes in Appendix A.1, while Appendix A.2 introduces supermartingales for the unfamiliar reader.

**Why use e-processes over p-processes?** Wang and Ramdas [44] describe a multitude of advantages outside of the bandit setting; these advantages also apply to the bandit setting but we do not redescribe them here for brevity. However, we will describe multiple ways in which using e-variables instead of p-variables as a measure of evidence in the bandit setting allows for both better flexibility and sample complexity of the algorithm. While this question has been the focus of a recent line of work for hypothesis tests in general [33, 41, 14, 44], we will explore how the properties of e-variables allow us to consider novel bandit setups and algorithms. In particular, e-variables allow us to be robust to arbitrary dependencies between statistics computed for each arm without additional correction. Further, we explore how e-processes can be merged under different conditions in Appendix D.3 to facilitate incorporation of existing evidence and cooperation between multiple agents and present concrete ways to construct e-processes in Appendices B.3 and F.

Since any non-trivial bandit algorithm will base its sampling choice on the rewards attained so far for every arm, average rewards of each arm are biased and dependent on each other in complex ways even if the algorithm is stopped at a fixed time [27, 35, 36, 37]. Even under a non-adaptive uniform sampling rule, an adaptive stopping rule can induce complex dependencies between reward statistics of each

arm. When using both adaptive sampling and stopping, the dependence effects are only compounded. Nevertheless, e-variable based algorithms enable us to prove FDR guarantees without assumptions on the sampling method. In contrast, procedures involving p-variables, such as the ones used in JJ, require the test level of $\alpha$ to be corrected by a factor of at least $\log(1/\alpha)$ when rewards are independent across arms, and a factor of $\log k$ otherwise. We expand on this in Section 2.2.

## 2.2   Multiple testing procedures with FDR control

We now introduce two multiple testing procedures that output a rejection set with provable FDR control. We will first describe the guarantees provided by the BH procedure [6], a classic multiple testing procedure that operates on p-variables. Then, we will describe e-BH, the e-variable analog of BH. Our key message in this section is that classical BH will have looser or tighter control of the FDR based upon the dependence structure of the p-variables it is operating on. On the other hand, e-BH provides a consistent guarantee on the FDR even when the e-variables are arbitrarily dependent. Both procedures take an input parameter $\alpha \in (0,1)$ that controls the degree of FDR guarantee (i.e. test level).

**Benjamini-Hochberg (BH) requires corrections for dependence and self-consistency.** A set $\mathcal{S}$ of p-values is called *p-self-consistent* [9] at level $\alpha$ iff:

$$\max_{i \in \mathcal{S}} p_i \leq \frac{|\mathcal{S}|\alpha}{k}. \tag{1}$$

The BH procedure with input $p_1,...,p_k$ outputs the largest p-self-consistent set w.r.t. the input, which we denote $\mathrm{BH}[\alpha](p_1,...,p_k)$. We must also define a condition on the joint distribution of $P_1,...,P_k$, which is called positive regression dependence on subset (PRDS). A formal definition is provided in Benjamini and Yekutieli [7], and it is sufficient for our purposes to think of this condition as positive dependence between $P_1,...,P_k$, with independence being a special case. Now, we describe the FDR control of the BH procedure.

**Fact 1** (BH FDR control.   Benjamini and Hochberg [6], Benjamini and Yekutieli [7]). *Let* $\mathcal{S} = \mathrm{BH}[\alpha](p_1,...,p_k)$. *If* $P_1,...P_k$ *are PRDS, then* $\mathrm{FDR}(\mathcal{S}) \leq \alpha$. *Otherwise, under arbitrary dependence amongst* $P_1,...P_k$, *the BH procedure ensures* $\mathrm{FDR}(\mathcal{S}) \leq \alpha \ell_k$, *where* $\ell_k \equiv \sum_{i=1}^k 1/k \approx \log k$.

Thus, in the case of arbitrary dependence, the FDR control of BH is larger by a factor of $\ell_k \approx \log k$. A larger FDR guarantee is provided for arbitrary p-self-consistent sets.

**Fact 2** (P-self-consistent FDR control. Su [38], Blanchard and Roquain [9], Wang and Ramdas [44]). *If* $\mathcal{S}$ *is p-self-consistent at level* $\alpha$ *and* $P_1,...,P_k$ *satisfy PRDS,* [1] *then* $\mathrm{FDR}(\mathcal{S}) \leq \alpha(1 + \log(1/\alpha))$. *Otherwise, when there is arbitrary dependence among* $P_1,...,P_k$, $\mathrm{FDR}(\mathcal{S}) \leq \alpha \ell_k$ *(consequence of Proposition 2.7 from Blanchard and Roquain [9] and Proposition 5.2 from Wang and Ramdas [44]).*

These two facts do not imply each other; the BH procedure outputs the largest self-consistent set and has a stronger or equivalent error guarantee under either type of dependence. While it may seem like we should always use BH and the guarantee from Fact 1 to form a rejection set, we elaborate in Section 3.1 on how we can use Fact 2 to provide FDR control for BH when the p-variables are not necessarily PRDS, and in settings where we may not directly use BH.

**e-BH needs no correction for dependence or self-consistency.** The e-BH procedure created by Wang and Ramdas [44] uses e-variables instead of p-variables and proceeds similarly to the BH procedure. In this case, let $e_1,...,e_k$ be the realized e-values for a set of e-variables $E_1,...,E_k$. Define $e_{[i]}$ to be the $i$th largest e-value for $i \in [k]$. A set $\mathcal{S}$ is *e-self-consistent* at level $\alpha$ iff $\mathcal{S}$ satisfies the following:

$$\min_{i \in \mathcal{S}} e_i \geq \frac{k}{\alpha |\mathcal{S}|}. \tag{2}$$

The e-BH procedure outputs the largest e-self-consistent set, which we denote by $\mathrm{eBH}[\alpha](e_1,...,e_k)$. For e-variables, the same guarantee applies for all e-self-consistent sets and under all dependence structures.

**Fact 3** (E-variable self-consistency FDR control. Wang and Ramdas [44]). *If* $\mathcal{S}$ *is e-self-consistent at level* $\alpha$, *then* $\mathrm{FDR}(\mathcal{S}) \leq \alpha$ *regardless of the dependence structure.*

All FDR bounds discussed in Facts 1 to 3 are optimal, in the sense that there exist e-variable/p-variable distributions with an FDR that is arbitrarily close or equivalent to the stated bound. Consequently, e-variables are more advantageous, since their FDR control does not change under different types of

---

[1]Su [38] technically employs a *slightly* weaker condition which implies PRDS, and refers to self-consistency as "compliance" (or, better said, compliance is a special case of self-consistency).

dependence as opposed to the factor of $1+\log(1/\alpha)$ or $\log k$ p-variables pay on the FDR for different settings.

In the case where p-variables can only be constructed as $P=1/E$, where $E$ is an e-variable, the rejection sets output by BH and e-BH are identical. However, the e-self-consistency guarantee in Fact 3 provides identical or tighter FDR control than the BH procedure guarantee in Fact 1 or p-self-consistency guarantee in Fact 2. Thus, e-variables and e-BH offer a degree of robustness against arbitrary dependence, since any algorithm using e-BH does not have to adjust $\alpha$ to guarantee the same level of $\mathrm{FDR}(\mathcal{S})\leq\delta$ for a fixed $\delta$ under different dependence structures. We now provide a meta-algorithm that utilizes p-self-consistency and e-self-consistency to guarantee FDR control in the bandit setting.

## 3   Decoupling exploration and evidence: a unified framework

We propose a framework for bandit algorithms that separates each algorithm into an **exploration** component and an **evidence** component; Algorithm 1 specifies a meta-algorithm combining the two.

---

**Algorithm 1:** A meta-algorithm for bandit multiple testing that decouples exploration and evidence. The evidence component can track p-processes or e-processes for each arm and use BH or e-BH.

---

**Input:** Exploration component $(\mathcal{A}_t)$, stopping rule $\tau^*$, Let $(p_{1,t}),...,(p_{k,t})$ and $(e_{1,t}),...,(e_{k,t})$
      denote the realized values of p-processes and e-processes, respectively. Let the desired
      level of FDR control be $\delta\in(0,1)$. Let $\delta'$ be the correction of $\delta$ for BH based upon the
      dependencies of $X_{1,t},...,X_{k,t}$. Set $D_0=\emptyset$.

**for** $t$ *in* $1...$ **do**
    $\mathcal{I}_t:=\mathcal{A}_t(D_{t-1})\subseteq[k]$
    Obtain rewards for each $i\in\mathcal{I}_t$, and update data $D_t:=D_{t-1}\cup\{(i,t,X_{i,t}):i\in\mathcal{I}_t\}$.
    Update e-process or p-process for each queried arm (summarizing evidence against each null).
    $\mathcal{S}_t:=\begin{cases}\mathrm{BH}[\delta'](p_{1,t},...,p_{k,t}) \text{ or arbitrary p-self-consistent set} & \text{if using p-variables}\\ \mathrm{eBH}[\delta](e_{1,t},...,e_{k,t}) \text{ or arbitrary e-self-consistent set} & \text{if using e-variables}\end{cases}$
    **if** $\tau^*=t$ **then** stop and **return** $\mathcal{S}_t$;
**end**

---

**Exploration component.** This is a sequence of functions $(\mathcal{A}_t)$, where $\mathcal{A}_t:\mathcal{F}_{t-1}\mapsto\mathcal{K}$ specifies the queried arms $\mathcal{I}_t:=\mathcal{A}_t(D_{t-1})$, and $D_t:=\{(i,j,X_{i,j}):j\leq t,i\in\mathcal{I}_j\}$ is the observed data. $\mathcal{A}_t$ is "non-adaptive" if it does not depend on the data, but only on some external randomness $U$. Regardless of how the exploration component $(\mathcal{A}_t)$ is constructed, our framework guarantees that $\mathrm{FDR}(\mathcal{S})\leq\delta$ for a fixed $\delta$. Similarly, $\tau^*$ is adaptive if it depends on the data, and is not determined purely by $U$.

**Evidence component.** The FDR control provided by Algorithm 1 is solely due to the formulation of the candidate rejection set, $\mathcal{S}_t\subseteq[k]$, at each time $t\in\mathbb{N}$ in the evidence component. This construction is completely separate from $(\mathcal{A}_t)$. Critically, $(\mathcal{S}_t)$ satisfies $\mathrm{FDR}(\mathcal{S}_\tau)\leq\delta$ for any stopping time $\tau\in\mathcal{T}$. This is accomplished by applying BH or e-BH to p-processes or e-processes, respectively. At stopping time $\tau$, $P_{i,\tau}$ is a p-variable when $(P_{i,t})$ is a p-process, and similarly $E_{i,\tau}$ is an e-variable when $(E_{i,t})$ is an e-process. Thus, $\mathcal{S}_\tau$ is the result of applying BH to p-variables or e-BH to e-variables.

Consequently, the aforementioned framework allows us to guarantee $\sup_{\tau\in\mathcal{T}}\mathrm{FDR}(\mathcal{S}_\tau)\leq\delta$ in a way that is agnostic to the exploration component. For completeness, we do discuss some generic exploration strategies in Appendix C. In the next section, we will formalize these guarantees and discuss the benefits afforded by using e-variables and e-BH in this framework instead of p-variables and BH.

### 3.1   FDR **control under different dependence structures**

In the general combinatorial bandit setting, different dependence structures affect the choice of $\delta'$ that ensures FDR control at $\delta$ in the p-variable and BH case. Table 1 summarizes the guarantees and choices of $\delta'$ for each type of dependence. Prior work on hypothesis testing in the bandit setting by JJ has only considered the non-combinatorial bandit case where $X_{1,t},...,X_{k,t}$ are independent. Critically, JJ employ BH and p-variables in their algorithm, and the FDR guarantee of BH changes based on the dependencies between reward distributions. On the other hand, choosing $\alpha=\delta$ for e-BH is sufficient to guarantee FDR control at level $\delta$ for any type of dependence between e-variables, but only sufficient for BH in the non-adaptive, PRDS $X_{1,t},...,X_{k,t}$ setting. We show that there is a wide range of dependence structures that require different degrees of correction for BH. Specifically, we will set an appropriate choice of $\delta'$ in each of these situations such that Algorithm 1 with p-variables can ensure FDR control level $\delta$. We include proofs of all results in this section in Appendix B.1.

Table 1: FDR control for BH, and the $\delta'$ to ensure $\delta$ control of FDR in Algorithm 1 under different dependence structures and adaptivity of $(\mathcal{A}_t)$. Adaptivity and arbitrary dependence both require extra correction for BH, but *any e-self-consistent procedure provides* $\mathrm{FDR}(\mathcal{S}) \leq \alpha$ *in all settings in the table.*

| **Adaptivity** of $(\mathcal{A}_t)$ and $\tau^*$ | **Dependence of** $X_{1,t},...,X_{k,t}$ | |
| --- | --- | --- |
| | *independent* | *arbitrarily dependent* |
| *non-adaptive* | $\mathrm{FDR}(\mathcal{S}) \leq \alpha$ 
 $\delta' = \delta$ | $\mathrm{FDR}(\mathcal{S}) \leq \alpha \log k$ 
 $\delta' = \delta/\log k$ (Prop. 2) |
| *adaptive* | $\mathrm{FDR}(\mathcal{S}) \leq \alpha((1+\log(1/\alpha)) \wedge \log k)$ 
 $\delta' = c_\delta \vee (\delta/\log k)$ (Prop. 1) | |
| Any **e-self-consistent procedure** ensures $\mathrm{FDR}(\mathcal{S}) \leq \alpha$ in all settings and sets $\alpha = \delta$. | | |

**Adaptive** $(\mathcal{A}_t)$ **and independent** $X_{1,t},...,X_{k,t}$**.** JJ consider this case in the non-combinatorial bandit setting, but their insights and techniques also can be extended to the combinatorial setting. We give a sketch of their proof here, and produce the full proof in Appendix B.1. In the language of self-consistency (not explicitly used in JJ), JJ make the key insight that *running BH on the p-variables for each arm produces a rejection set that is actually p-self-consistent with a different set of independent p-variables.* Define $P_1^*,...,P_k^*$, where $P_i^* = \inf_{t \in \mathbb{N}} P_{i,t}$ for each $i \in [k]$ i.e. each arm's p-variable in the infinite sample limit. Since $(P_{i,t})$ is a p-process for each arm $i \in [k]$, the corresponding $P_i^*$ is a p-variable (Proposition 6 in Appendix A.1). Further, $P_1^*,...,P_k^*$ are independent because $X_{1,t},...,X_{k,t}$ are independent. By definition of $P_1^*,...,P_k^*$, $p_i^* \leq p_{i,t}$ for any $i \in [k]$ and any $t \in \mathbb{N}$. Thus, $\mathcal{S}_{\tau^*}$ is p-self-consistent w.r.t. $p_1^*,...,p_k^*$, and has its FDR bounded by $\alpha(1+\log(1/\alpha))$ due to Fact 2. At the same time, the arbitrary dependence guarantee from Fact 1 still applies. Combining these facts, we achieve the following guarantee:

**Proposition 1.** *When $(\mathcal{A}_t)$ is adaptive and $X_{1,t}, ... , X_{k,t}$ are independent, Algorithm 1 with p-processes and an arbitrary p-self-consistent set guarantees* $\sup_{\tau \in \mathcal{T}} \mathrm{FDR}(\mathcal{S}_\tau) \leq \delta$ *if* $\delta' \leq c_\delta \vee \delta/\ell_k$, *where for any* $\delta \in (0,1)$, *define* $c_\delta \leq \delta$ *as the solution to* $c_\delta(1+\log(1/c_\delta)) = \delta$.

Note that Proposition 1 is valid for any p-self-consistent set since p-self-consistency is the only property required of the output set to prove the result. JJ prove a similar bound to Proposition 1. However, they used a larger FDR bound for p-self-consistent sets with worse constants (which was subsequently improved by Su [38] as presented earlier), and they only considered the non-combinatorial case. Proposition 1 uses an optimal bound on p-self-consistent sets from Fact 2, and is valid in our combinatorial bandit setup.

**Adaptive** $(\mathcal{A}_t)$ **and arbitrarily dependent** $X_{1,t},...,X_{k,t}$**.** In the general combinatorial bandit setting, where the algorithm chooses a subset of arms or "superarm" at each time to jointly sample from, we will have multiple samples from multiple arms in the same time step, and $X_{1,t},...,X_{k,t}$ can be arbitrarily dependent. Consequently, the p-variables corresponding to each arm can also be arbitrarily dependent. For example, a superarm could consist of all arms, and the sampling rule could be to just sample this superarm that encompasses all arms. Then, the p-variable distribution would directly depend on the reward distribution of the arms. Thus, we can provide the following guarantee by Fact 1 when using p-variables as a result of Fact 3.

**Proposition 2.** *When $(\mathcal{A}_t)$ is adaptive and $X_{1,t},...,X_{k,t}$ are dependent, Algorithm 1 with p-variables and BH guarantees* $\sup_{\tau \in \mathcal{T}} \mathrm{FDR}(\mathcal{S}_\tau) \leq \delta$ *if* $\delta' \leq \delta/\ell_k$.

Finally, consider a setting structured setting where we cannot output the rejection set of BH. Such a constraint often occurs in directed acyclic graph (DAG) settings where there is a hierarchy among hypotheses that restricts which rejection sets are allowed [30, 25]. Instead, we would like to output the largest self-consistent set that respects the structural constraints. By Fact 2, we get the following FDR control.

**Proposition 3.** *If $(\mathcal{A}_t)$ is adaptive and $X_{1,t},...,X_{k,t}$ are dependent, Algorithm 1 with p-variables that outputs an arbitrary p-self-consistent $\mathcal{S}_t$ guarantees* $\sup_{\tau \in \mathcal{T}} \mathrm{FDR}(\mathcal{S}_\tau) \leq \delta$ *if* $\delta' \leq c_\delta/\ell_k$.

We explore the structured setting with greater depth in Appendix D.2. Unlike p-variables, e-variables do not need correction in any of the aforementioned settings.

**Proposition 4.** *When $(\mathcal{A}_t)$ is adaptive and $X_{1,t},...,X_{k,t}$ are dependent, Algorithm 1 with e-variables, which runs e-BH at level $\delta$ or outputs a e-self-consistent set at level $\delta$, guarantees* $\sup_{\tau \in \mathcal{T}} \mathrm{FDR}(\mathcal{S}_\tau) \leq \delta$.

Thus, running e-BH (or any e-self-consistent procedure) at level $\delta$ is valid for any choice of $(\mathcal{A}_t)$ and type of dependence. Now, we give an example where $X_{1,t},...,X_{k,t}$ might be arbitrarily dependent.

## 3.2 Illustrative examples to demonstrate flexibility of the framework

Below, we briefly describe a set of nontrivial illustrative examples to showcase the flexibility of our framework. In most of the cases below, a p-process approach would have to correct for dependence and/or self-consistency in different case-specific ways, rendering it more conservative and requiring careful arguments to justify FDR control. However, working with our unified framework is easy, handling both self-consistency and dependence issues in the same breath and without any changes to the algorithm or analysis. The data scientist can focus on designing powerful e-processes *for each arm separately* and let the modular framework correct for the multiplicity aspect.

**Example: sampling nodes on a graph.** A scenario where $X_{1,t}, \dots, X_{k,t}$ may naturally have dependence is when each arm corresponds to a node on a graph. The superarms in this situation could be defined w.r.t. to a graph constraint e.g. "two nodes connected by an edge" or "a node and its neighbors". Graph bandits has been studied in the regret setting [26] and have many real world applications [39]. We could imagine a scenario where low power sensors in a sensor network can only communicate locally. A centralized algorithm is tasked with querying the sensors to find those with high activity. A sensor may only provide activity information about itself and nearby sensors, and this data can be arbitrarily dependent across the sensors. Figure 1 illustrates a superarm in this situation.

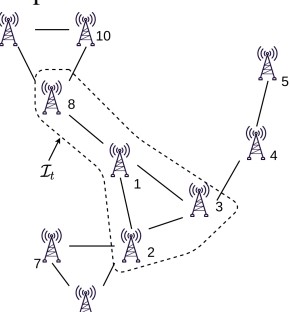

In such a setting, if Proposition 2 is used to guarantee $\text{FDR}(\mathcal{S}) \leq \delta$ with p-variables, it pays a $\log k$ correction, while Proposition 4 can guarantee e-variables need no correction. We simulate this setting in Appendix E.2, and show these differences empirically. We also discuss some other examples in the appendix that we will summarize here.

- **Multiple agents** (Appendix D.3): Consider the setting where multiple agents are operating on the same bandit, and we want to aggregate the evidence for rejection across agents. For e-processes, we present an algorithm for merging e-values that maintains FDR control.

- **Structured rejection sets** (Appendix D.2): We illustrate the difference between self-consistency guarantees for p-variables and e-variables when a DAG hierarchy is imposed upon the hypotheses.

Figure 1: A superarm consists of a node and all its neighbors. The dotted line captures $\mathcal{I}_t$, the superarm around node 1.

- **Multi-arm hypotheses** (Appendix D.4) A hypothesis may concern the reward distributions of multiple arms e.g. are the means of two different arms equivalent? We provide FDR guarantees even when hypotheses and arms are not matched one-to-one.

- **Streaming data setting** (Appendix D.1) Our methods also naturally extend to the streaming setting when the algorithm views the rewards of every at each time step.

Now that we have shown FDR is controlled using e-variables in a way that is robust to the underlying dependence structure, we analyze the sample complexity of achieving a high TPR using e-variables when the rewards are independent and sub-Gaussian.

## 4 E-process sample complexity guarantees for sub-Gaussian arms

We provide sample complexity guarantees for the sub-Gaussian setting that has been the focus of existing methodology by JJ in bandit multiple testing. We explicitly define e-processes and an exploration component $(\mathcal{A}_t)$ that will have sample complexity bounds matching those of the algorithm in JJ, which uses p-variables. Specifically, we will consider the standard bandit setting where $|\mathcal{I}_t| = 1$ and $\nu_i$ is 1-sub-Gaussian for each $i \in [k]$. Denote the means of each arm $i \in [k]$ as $\mu_i = \mathbb{E}[X_{i,t}]$ for all $t \in \mathbb{N}$. The goal is to find many arms where $\mu_i > \mu_0$, where we set $\mu_0 = 0$ to be the mean of a reward distribution under the null hypothesis. Thus, we define $\mathcal{H}_0 = \{i \in [k] : \mu_i \leq \mu_0\}$ and $\mathcal{H}_1 = \{i \in [k] : \mu_i > \mu_0\}$. Our framework ensures that $\text{FDR}(\mathcal{S}) \leq \delta$, and we also want to achieve $\text{TPR}(\mathcal{S}) \geq 1 - \delta$ with small sample complexity. Proofs of the results from this section are in Appendices B.2 and B.3. As an aside, we also discuss what hypotheses we can test when the reward distribution is not necessarily independent across $t \in \mathbb{N}$, but the conditional distribution of the rewards still satisfy certain sub-Gaussian guarantees in Appendix G.

Our e-process of choice is the **discrete mixture e-process** from Howard et al. [16]:

**Proposition 5.** $E_{i,t}^{\text{DM}}$ is an e-process when $\nu_i$ is 1-sub-Gaussian and $i \in \mathcal{H}_0$.

Denote $\Delta_i \equiv \mu_i - \mu_0$ for $i \in \mathcal{H}_1$ and $\Delta \equiv \min_{i \in \mathcal{H}_1} \Delta_i$. When $i \in \mathcal{H}_0$, let $\Delta_i \equiv \min_{j \in \mathcal{H}_1} \mu_j - \mu_0 = \Delta + (\mu_i - \mu_0)$. First, we recall a time-uniform bound on the sample mean $\widehat{\mu}_t$.

$$E_{i,t}^{\mathrm{DM}}(\mu_0) := \sum_{\ell=0}^{\infty} w_\ell \exp\left(\sum_{j=1}^{T_i(t)} \lambda_\ell(X_{i,t_i(j)} - \mu_0) - \lambda_\ell^2/2\right), \quad (3a)$$

$$\text{where } \lambda_\ell := \frac{1}{e^{\ell+5/2}} \text{ and } w_\ell := \frac{2(e-1)}{e(\ell+2)^2} \text{ for } \ell \in \mathbb{N}_0. \quad (3b)$$

**Fact 4** (JJ, Kaufmann et al. [23], Howard et al. [16]). *Let $X_1, X_2, ...$ be i.i.d. draws from a 1-sub-Gaussian distribution with mean $\mu$. Consider the boundaries defined in* (4). *Let $\varphi$ be one of these boundaries. Then,* $\mathbb{P}(\exists t \in \mathbb{N} : |\widehat{\mu}_t - \mu| > \varphi(t,\delta)) \leq \delta$ *for any $\delta \in (0,1)$ if $\varphi \in \{\varphi^0, \varphi^{\mathrm{IS}}\}$ and any $\delta \in (0,0.1]$ if $\varphi = \varphi^{\mathrm{JJ}}$.*

We will use $\varphi$ to refer to an arbitrary boundary from Fact 4. All of the $\varphi$ are time-uniform boundaries that yield confidence sequences for the mean. Note that $\varphi^0$ is generally larger than the other boundaries, so we use $\varphi^0$ as the default boundary in our

$$\varphi^0(t,\delta) := \sqrt{\frac{4\log(\log_2(2t)/\delta)}{t}}, \quad (4a)$$

$$\varphi^{\mathrm{JJ}}(t,\delta) := \sqrt{\frac{2\log(1/\delta) + 6\log\log(1/\delta) + 3\log(\log(et/2))}{t}}, \quad (4b)$$

$$\varphi^{\mathrm{IS}}(t,\delta) := \sqrt{\frac{2.89\log\log(2.041t) + 2.065\log\left(\frac{4.983}{\delta}\right)}{t}}. \quad (4c)$$

proofs, and we explore how different choices of $\varphi$ affect empirical performance in Appendix E.1. Now, we can define the algorithm from JJ in (5), which consists of an exploration policy based on an upper confidence bound (UCB) of the mean reward (specified by a singleton set $\mathcal{I}_t = \{I_t\}$) and a p-variable derived from Fact 4.

In (5a), we denote the sample mean at time $t$ of each arm $i \in [k]$ by $\widehat{\mu}_{i,t}$. Let $f \lesssim g$ denote $f$ asymptotically dominates $g$ i.e. there exist $c > 0$ that is independent of the problem parameters such that $f \leq cg$. JJ prove the following sample complexity guarantee for their algorithm.

**Fact 5** (From JJ). *Let $(\mathcal{A}_t)$ output $\mathcal{I}_t = \{I_t\}$, and let $I_t$ and $P_{i,t}$ be specified by Alg. 5 with $\varphi = \varphi^0$. Then, Algorithm 1 will always guarantee $\sup_{\tau \in \mathcal{T}} \mathrm{FDR}(\mathcal{S}_\tau) \leq \delta$. With at least $1-\delta$ probability, there will exist $T \lesssim \left(\sum_{i=1}^{k} \Delta_i^{-2}\log\log\Delta_i^{-2} + \Delta_i^{-2}\log(k/\delta)\right) \wedge k\Delta^{-2}\log(\log(\Delta^{-2})/\delta)$ such that $\mathrm{TPR}(\mathcal{S}_t) \geq 1-\delta$ for all $t \geq T$.*

We show that we can match the sample complexity bounds of Fact 5 with e-variables.

$$I_t = \operatorname*{argmax}_{i \in [k] \setminus \mathcal{S}_{t-1}} \widehat{\mu}_{i,t-1} + \varphi(T_i(t-1),\delta), \quad (5a)$$

$$P_{i,t} \equiv \inf\{\rho \in [0,1] : |\widehat{\mu}_{i,t} - \mu_0| > \varphi(t,\rho)\}. \quad (5b)$$

**Theorem 1.** *Let $\nu_i$ be 1-sub-Gaussian for $i \in [k]$. Set $(\mathcal{A}_t)$ so $\mathcal{A}_t$ outputs $\{I_t\}$ from (5a) for all $t \in \mathbb{N}$ and $E_{i,t}$ to $E_{i,t}^{\mathrm{DM}}$. Algorithm 1 ensures $\sup_{\tau \in \mathcal{T}} \mathrm{FDR}(\mathcal{S}_\tau) \leq \delta$ and, with at least $1-\delta$ probability, there exists $T \lesssim \left(\sum_{i=1}^{k} \Delta_i^{-2}\log\log\Delta_i^{-2} + \Delta_i^{-2}\log(k/\delta)\right) \wedge k\Delta^{-2}\log(\log(\Delta^{-2})/\delta)$ such that $\mathrm{TPR}(\mathcal{S}_t) \geq 1-\delta$ for all $t \geq T$.*

In addition to matching theoretical guarantees, we show in the following section that e-variables and e-BH perform empirically as well or better than p-variables and BH through numerical simulations.

## 5 Numerical simulations

We perform simulations for the sub-Gaussian setting discussed in Section 4 to demonstrate that our version of Algorithm 1 using e-variables is empirically as efficient as the algorithm of JJ, which uses p-variables (code available here) . However, unlike JJ, our algorithm does not use a corrected level $\delta'$ based upon the dependence assumptions among $X_{1,t},...,X_{k,t}$ to guarantee FDR is controlled at level $\delta$. We explore additional simulations of combinatorial semi-bandit settings with dependent $X_{1,t},...,X_{k,t}$ in Appendix E that show the benefit of using e-variables over p-variables in our framework.

**Simulation setup** Let $\nu_i = \mathcal{N}(\mu_i,1)$ where $\mu_i = \mu_0 = 0$ if $i \in \mathcal{H}_0$ and $\mu_i = 1/2$ if $i \in \mathcal{H}_1$. We consider 3 setups, where we set the number of non-null hypotheses to be $|\mathcal{H}_1| = 2, \log k$, and $\sqrt{k}$, to see the effect of different magnitudes of non-null hypotheses on the sample complexity of each method. We set $\delta = 0.05$ and compare 4 different methods. We compare the same two different exploration components for both e-variables and p-variables. The first exploration component we consider is simply uniform sampling across each arm (Uni). The second is the UCB sampling strategy described in (5a). When using BH, our formulation for p-variables is (5b), which is the same as JJ. Like JJ, we set $\varphi = \varphi^{\mathrm{JJ}}$ in our simulations. When using e-BH, we set our e-variables to $E_{i,t}^{\mathrm{PM-H}} := \prod_{j=1}^{T_i(t)} \exp(\lambda_{i,t_i(j)}(X_{i,t_i(j)} - \mu_0) - \lambda_{i,t_i(j)}^2/2)$

with $\lambda_{i,t} = \sqrt{\frac{2\log(2/\alpha)}{T_i(t)\log(T_i(t)+1)}}$, which is the default choice of $\lambda_{i,t}$ suggested in Waudby-Smith and Ramdas [45]. We show that this is a valid e-process in Appendix F and maintains FDR control.

**Results** We plot the relative performance of each method to e-BH with UCB sampling in Figure 2. For uniform sampling, e-BH and e-variables seem to outperform BH and p-variables, although by a decreasing margin for more arms, especially in the case where $|\mathcal{H}_1| = \lfloor\sqrt{k}\rfloor$. For the UCB sampling algorithm, we see that e-variables and p-variables have relatively similar performance, with the gap narrowing as the number of arms increase as well. Thus, e-variables and e-BH empirically perform on par or better than p-variables with regards to sample complexity. This shows that using e-variables does not require any sacrifice in performance in simple cases where p-variables also work well. Further, e-variables do not require the same $\log k$ correction that p-variables need for situations where $X_{1,t},...,X_{k,t}$ are arbitrarily dependent to guarantee FDR control at the same level. Thus, e-variables are preferable to p-variables as they are more flexible w.r.t. assumptions.

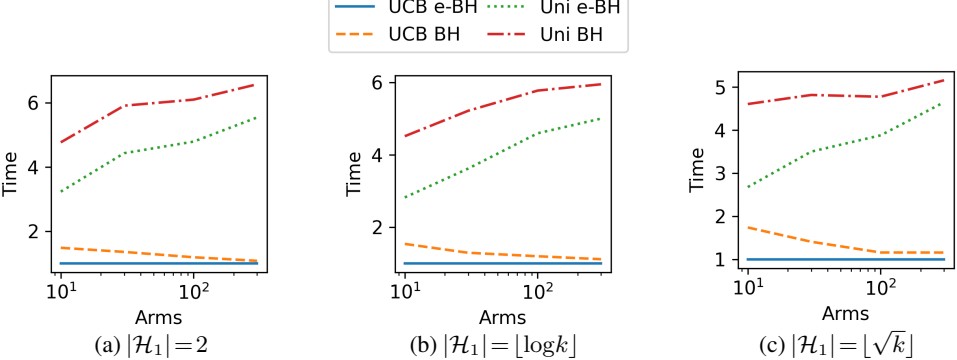

(a) $|\mathcal{H}_1| = 2$        (b) $|\mathcal{H}_1| = \lfloor\log k\rfloor$        (c) $|\mathcal{H}_1| = \lfloor\sqrt{k}\rfloor$

Figure 2: Relative comparison of time $t$ to obtain a rejection set, $\mathcal{S}_t$, that has a $\text{TPR}(\mathcal{S}_t) \geq 1-\delta$ and $\text{FDR}(\mathcal{S}_t) \leq \delta$ where $\delta = 0.05$. This plot compares e-BH vs. BH for both uniform (Uni) and UCB sampling over different numbers of arms (choices of $k$) and densities of non-null hypotheses (sizes of $\mathcal{H}_1$). Time is reported as a ratio to the time taken by UCB e-BH method. Note that the methods using e-variables perform on par or better than methods using p-variables for both sampling strategies.

## 6 Conclusion, limitations and broader impact

In this paper, we developed a unified framework for bandit multiple hypothesis testing. We demonstrated that applying the e-BH procedure to stopped e-processes guarantees FDR control without assumptions on the the dependency between $X_{1,t},...,X_{k,t}$, exploration strategy, stopping time of the algorithm, ability to query multiple arms, etc. In contrast, existing algorithms using BH and p-variables have FDR guarantees that vary with the problem setting and dependence structure among the p-variables. We argued that control of the FDR with p-variables can blow up by a factor of $\log k$, and any p-self-consistent algorithm must decrease its threshold for discovery correspondingly to maintain FDR control at the desired level. We provide more detailed explanations of these observations in Appendix D.2. In addition to demonstrating the generality of our meta-algorithm, we showed that in the standard sub-Gaussian reward setting, the instantiated algorithm matches the sample complexity bounds of the p-variable algorithm by JJ for achieving high TPR, and has better practical performance than JJ's algorithm, despite the fact that we improve JJ's guarantees by invoking the self-consistency results of Su [38].

The appendices have additional examples of problem settings and simulations that show the utility of e-processes and our general framework. In fact, we can address an even more general setting where the null hypotheses do not have a one-to-one correspondence with the arms; in other words, despite the queries being at the arm-level, the hypotheses being tested could combine arms (for example, comparing different arms). We also discuss the multi-agent setting where there could be multiple agents that operate the same bandit. We avoided these scenarios in the main paper for simplicity of exposition, since there were enough generalizations to describe in the simpler setup already.

The main limitation of the work is that it does not develop instance optimal sampling algorithms for multiple testing problem in the described settings with more complicated dependence structures; we believe this is a difficult open problem, requiring specialized techniques in each example. We do not foresee any negative societal impact of this work; it is aimed at reducing costs and improving reproducibility in scientific experimentation by controlling false discoveries in adaptive testing.

**Acknowledgments**   RW acknowledges funding from NSERC RGPIN-2018-03823 and RGPAS-2018-522590. AR acknowledges funding from NSF DMS 1916320 and ARL IoBT REIGN. Research reported in this paper was sponsored in part by the DEVCOM Army Research Laboratory under Cooperative Agreement W911NF-17-2-0196 (ARL IoBTCRA). The views and conclusions contained in this document are those of the authors and should not be interpreted as representing the official policies, either expressed or implied, of the Army Research Laboratory or the U.S. Government. The U.S. Government is authorized to reproduce and distribute reprints for Government purposes notwithstanding any copyright notation herein.

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
