# A Miscellaneous technicalities

Here, we collect some definitions and properties concerning p-process and supermartingales for the unfamiliar reader, and restate a technical lemma necessary for upcoming proofs.

**Lemma 1** (Lemma 8 from JJ). *Let $a \in \mathbb{R}_+^n$ be a $n$-dimensional vector with positive real entries, and for $i = 1,...,n$ let $Z_i$ be independent random variables where*

$$\mathbb{P}(Z_i \geq t) \leq \exp(-t/a_i).$$

*Then for any $\delta \in (0,1)$,*

$$\sum_{i=1}^n Z_i \leq 5\log(1/\delta) \sum_{i=1}^n a_i.$$

*occurs with at least probability $1 - \delta$.*

## A.1 Equivalence property of p-processes

We note the following equivalence proposition for p-processes. Lemma 3 in Howard et al. [16] and Lemmas 1 and 2 in Ramdas et al. [31] makes similar statements regarding sequential processes, but do not additionally characterize the behavior of the infimum of a p-process.

**Proposition 6.** *The following statements are equivalent for a discrete-time process $(P_t)_{t \geq 1}$:*

*(i) $(P_t)_{t \geq 1}$ is a p-process i.e. $\mathbb{P}(P_\tau \leq \alpha) \leq \alpha$ for all (possibly infinite) $\tau \in \mathcal{T}$ and all $\alpha \in (0,1)$;*

*(ii) $\mathbb{P}(P_\tau \leq \alpha) \leq \alpha$ for all finite $\tau \in \mathcal{T}$ and all $\alpha \in (0,1)$;*

*(iii) $\mathbb{P}(\exists t \geq 1 : P_t \leq \alpha) \leq \alpha$ for all $\alpha \in (0,1)$;*

*(iv) $\inf_{t \geq 1} P_t$ is superuniformly distributed (its distribution is stochastically larger than uniform).*

*Proof.* In what follows, let $\tau_\alpha \in \mathcal{T}$ be defined as $\tau_\alpha := \inf\{t \geq 1 : P_t \leq \alpha\}$, which is defined to be infinite if $P_t$ never drops below $\alpha$.

(i)$\Rightarrow$(ii) is trivial by definition.

(ii)$\Rightarrow$(iii): Fix $\alpha \in (0,1)$. By (ii), we have $\mathbb{P}(P_{\tau_\alpha \wedge n} \leq \alpha) \leq \alpha$ for all $n \geq 1$. It follows that

$$\mathbb{P}(\exists t \geq 1 : P_t \leq \alpha) = \lim_{n \to \infty} \mathbb{P}(\exists t \in \{1,...,n\} : P_t \leq \alpha) = \lim_{n \to \infty} \mathbb{P}(P_{\tau_\alpha \wedge n} \leq \alpha) \leq \alpha.$$

(iii)$\Rightarrow$(iv): For each $\epsilon > 0$, since $\inf_{t \geq 1} P_t \leq \alpha$ implies $\tau_{\alpha + \epsilon} < \infty$, we have

$$\mathbb{P}\left(\inf_{t \geq 1} P_t \leq \alpha\right) \leq \mathbb{P}(\tau_{\alpha + \epsilon} < \infty) \leq \alpha + \epsilon.$$

As $\epsilon > 0$ is arbitrary, we get $\mathbb{P}(\inf_{t \geq 1} P_t \leq \alpha) \leq \alpha$, i.e., $\inf_{t \geq 1} P_t$ is superuniformly distributed.

(iv)$\Rightarrow$(i): For any $\tau \in \mathcal{T}$ and $\alpha \in (0,1)$, since $P_\tau \geq \inf_{t \geq 1} P_t$, we have $\mathbb{P}(P_\tau \leq \alpha) \leq \mathbb{P}(\inf_{t \geq 1} P_t \leq \alpha) \leq \alpha$, thus showing that $(P_t)$ is a p-process. $\square$

As a direct consequence of Proposition 6, if $(P_t)$ is a p-process and $P_s$ is uniformly distributed on $[0,1]$ for some $s \geq 1$, then we have $\mathbb{P}(P_s \leq P_t) = 1$ for all $t \geq 1$, since $P_s$ is as small as $\min_{t \geq 1} P_t$. Therefore, if a p-process does not always take its minimum at a deterministic point $s$, then $P_s$ cannot be uniformly distributed on $[0,1]$. In other words, for all deterministic $t$, the random variables $P_t$ are, in general, not "precise" (i.e., uniform on $[0,1]$) p-variables, but conservative ones. In contrast, the random variables $E_t$ from an e-process $(E_t)$ are "precise" (i.e., have expectation 1) as soon as $(E_t)$ is a nonnegative martingale starting at 1.

## A.2 Nonnegative supermartingales

A real-valued process $(M_t)_{t \geq 0}$ is a supermartingale w.r.t. a filtration $(\mathcal{F}_t)$ if it satisfies:

$$\mathbb{E}[M_t | \mathcal{F}_{t-1}] \leq M_{t-1} \text{ for } t \in \mathbb{N}. \tag{6}$$

For nonnegative supermartingales, we typically assume $M_0 = 1$ for simplicity; they possess two useful properties. The first is the optional stopping theorem.

**Fact 6** (Optional stopping theorem. Durrett [13], Ramdas et al. [31])**.** *Let $(M_t)$ be a nonnegative supermartingale w.r.t. $(\mathcal{F}_t)$. Then, for any stopping time $\tau \in \mathcal{T}$:*

$$\mathbb{E}[M_\tau] \leq M_0.$$

The second is Ville's inequality.

**Fact 7** (Ville's inequality)**.** *Let $(M_t)$ be a nonnegative supermartingale w.r.t. $(\mathcal{F}_t)$. Let $s \in \mathbb{R}^+$ be a number in the positive reals.*

$$\mathbb{P}(\exists t \in \mathbb{N} : M_t \geq s) \leq \frac{M_0}{s}.$$

## B  Proofs

### B.1  Proofs of results in Section 3.1

The proofs of Propositions 1 to 4 all follow from the application of one of Facts 1 to 3.

First, we note that $(P_{1,t}), ..., (P_{k,t})$ being p-processes implies that $P_{1,t}, ..., P_{k,t}$ are p-variables for all $t \in \mathbb{N}$. Thus, for any choice of stopping time $\tau^* \in \mathcal{T}$ for the algorithm, $P_{1,\tau^*}, ..., P_{k,\tau^*}$ are p-variables.

Consequently, Proposition 2 for the adaptive and dependent p-variables case and Proposition 3 for the adaptive and dependent p-variables with constrained rejection sets case follow from Fact 1 and Fact 2, respectively.

Similarly, we note that $E_{1,\tau^*}, ... E_{k,\tau^*}$ are e-variables, since $(E_{1,t}), ..., (E_{k,t})$ are e-processes. As a result, Proposition 4 follows from Fact 3.

Now, we prove Proposition 1 in a slightly different manner than JJ, using the notion of self-consistency.

*Proof of Proposition 1.* Consider an arbitrary $i \in [k]$. Recall that each $P_{i,t}$ is determined only by $X_{i,t_i(1)}, ..., X_{i,t_i(T(i))}$. By independence of $X_{i,t}$ across $i \in [k]$ and $t \in \mathbb{N}$, we rename $X_{i,t_i(j)}$ as $X_{i,j}$, since they are identically distributed. Thus, $P_{i,t}$ is now constructed from $X_{i,1}, ..., X_{i,t}$. We perform this transformation so we can consider $P_{i,t}$ in the infinite-sample limit. Under our renaming, $\mathcal{S}_{\tau^*}$ is the output of running BH on $P_{1,T_1(\tau^*)}, ..., P_{k,T_k(\tau^*)}$.

Define $P_i^* := \inf_{t \geq 1} P_{i,t}$. Note that $P_i^*$ are independent p-variables across $i \in [k]$ by Proposition 6 since $X_{1,t}, ..., X_{k,t}$ are independent for all $t \in \mathbb{N}$ and $(P_{1,t}), ..., (P_{k,t})$ are p-processes. We can derive self-consistency w.r.t. $P_i^*$ as follows.

$$\max_{i \in \mathcal{S}_{\tau^*}} P_i^* \leq \max_{i \in \mathcal{S}_{\tau^*}} P_{i,T_i(\tau^*)} \qquad \text{def. of } P_i^*$$

$$\leq \frac{|\mathcal{S}_{\tau^*}|\delta'}{k}. \qquad \text{p-self-consistency of } \mathcal{S}_{\tau^*}$$

Combined with Fact 2, we can show that $\mathrm{FDR}(\mathcal{S}_{\tau^*}) \leq \delta' \log(1 + \log(1/\delta'))$.

Separately, we can also apply the FDR guarantee on the output of BH on arbitrarily dependent p-variables from Fact 1. Consequently, we can guarantee $\mathrm{FDR}(\mathcal{S}_{\tau^*}) \leq \delta' \log(1 + \log(1/\delta')) \wedge \delta' \log k$. Thus, our choice of $\delta'$ implies $\mathrm{FDR}(\mathcal{S}_{\tau^*}) \leq \delta$, which is our desired result. $\qquad \square$

### B.2  Proof of Proposition 5

Howard et al. [16] actually specifiy a more general form for $\lambda_\ell$ and $w_\ell$ for the discrete mixture e-process, $E_{i,t}^{\mathrm{DM}}$. Let $f$ be a probability density over $(0, \lambda^{\max}]$ and nonincreasing over that interval, $\overline{\lambda} \in \mathbb{R}^+$ satisfy $\overline{\lambda} \leq \lambda^{\max}$, and $\eta > 1$ be a step size. Howard et al. [15] define $\lambda_\ell, w_\ell$ as follows:

$$\lambda_\ell := \frac{\overline{\lambda}}{\eta^{\ell + 1/2}} \text{ and } w_k := \frac{\overline{\lambda}(\eta - 1)f(\lambda_\ell \sqrt{\eta})}{\eta^{\ell + 1}}. \qquad (7)$$

Let,

$$f_s^{\mathrm{LIL}} := \frac{(s-1)s^{s-1}\mathbf{I}\{0 \leq \lambda \leq 1/e^s\}}{\lambda \log^s \lambda^{-1}},$$

for any $s > 1$. We will now connect these definitions to (3b). Set $\overline{\lambda} = 1/e$, $\eta = e$, and $f = f_2^{\text{LIL}}$. Then,

$$
\lambda_\ell = \frac{1}{e^{\ell+3/2}},
$$

$$
w_\ell = \frac{\frac{1}{e}(e-1)f_2^{\text{LIL}}\left(\frac{1}{e^{\ell+3/2}} \cdot \sqrt{e}\right)}{e^{\ell+1}}
$$

$$
= \frac{(e-1)f_2^{\text{LIL}}\left(\frac{1}{e^{\ell+1}}\right)}{e^{\ell+2}}
$$

$$
= \frac{(e-1)\frac{2\mathbf{I}\{\ell \geq 1\}}{\frac{1}{e^{\ell+1}}\log^2(e^{\ell+1})}}{e^{\ell+2}}
$$

$$
= \frac{2(e-1)\mathbf{I}\{\ell \geq 1\}}{(\frac{1}{e^{\ell+1}})(e^{\ell+2})\log^2(e^{\ell+1})}
$$

$$
= \frac{2(e-1)\mathbf{I}\{\ell \geq 1\}}{e(\ell+1)^2}.
$$

By reindexing $\ell$, we can redefine the variables as follows:

$$
\lambda_\ell = \frac{1}{e^{\ell+5/2}} \text{ and } w_\ell = \frac{2(e-1)}{e(\ell+2)^2}.
$$

To prove Proposition 5, we prove the following more general proposition which is derived from existing results in Howard et al. [16].

**Proposition 7** (Derived from equations (49) and (82) of Howard et al. [16]). *Let,*

$$
E_{i,t} := \sum_{\ell=0}^{\infty} w_\ell \exp\left(\sum_{j=1}^{T_i(t)} \lambda_\ell(X_{i,t_i(j)} - \mu_0) - \frac{\lambda_\ell^2}{2}\right).
$$

*If $\sum_{\ell=0}^{\infty} w_\ell \leq 1$, then $(E_{i,t})$ is a nonnegative supermartingale, and consequently an e-process, if the conditional distribution $X_{i,t} \mid \mathcal{F}_{t-1}$ is 1-sub-Gaussian and $\mathbb{E}[X_{i,t} \mid \mathcal{F}_{t-1}] \leq \mu_0$ for all $t \in \mathbb{N}$.*

*Proof.* Let

$$
M_{i,t}^\lambda := \exp\left(\sum_{j=1}^{T_i(t)} \lambda(X_{i,t_i(j)} - \mu_0) - \frac{\lambda^2}{2}\right),
$$

where $\lambda \in \mathbb{R}$. $(M_{i,t})$ is a nonnegative supermartingale because of the sub-Gaussian and bounded conditional mean assumptions on $X_{i,t}$. Let, $w_{\text{sum}} = \sum_{\ell=0}^{\infty} w_\ell$. Now, we show that $E_{i,t}$ is a supermartingale:

$$
\mathbb{E}[E_{i,t} \mid \mathcal{F}_{t-1}] = \mathbb{E}\left[\sum_{\ell=0}^{\infty} w_\ell M_{i,t}^{\lambda_\ell} \mid \mathcal{F}_{t-1}\right]
$$

$$
= \sum_{\ell=0}^{\infty} w_\ell \mathbb{E}\left[M_{i,t}^{\lambda_\ell} \mid \mathcal{F}_{t-1}\right]
$$

$$
\leq \sum_{\ell=0}^{\infty} w_\ell M_{i,t-1}^{\lambda_\ell}
$$

$$
= E_{i,t-1}.
$$

The sole inequality is by the supermartingale property of $(M_{i,t}^{\lambda_\ell})$. Thus, we have shown our desired result. $\qquad \square$

### B.3 Proof of Theorem 1

We follow a similar path as the sample complexity proof (i.e. Theorem 2) from JJ for Theorem 1. Our goal is to show that we reject the following set with at least $1-\delta$ probability:

$$\mathcal{R} = \{i \in \mathcal{H}_1 : \widehat{\mu}_{i,t} + \varphi(t,\delta) \geq \mu_i \text{ for all } t \in \mathbb{N}\}. \tag{8}$$

**Lemma 2.** $\mathbb{E}[|\mathcal{R}|] \geq (1-\delta)|\mathcal{H}_1|$.

*Proof.* We have the following:

$$\begin{aligned}
\mathbb{E}[|\mathcal{R}|] &= \sum_{i \in \mathcal{H}_1} \mathbb{P}(\widehat{\mu}_{i,t} + \varphi(t,\delta) \geq \mu_i) \\
&\geq \sum_{i \in \mathcal{H}_1} \mathbb{P}(|\widehat{\mu}_{i,t} - \mu_i| \leq \varphi(t,\delta)) \\
&\geq (1-\delta)|\mathcal{H}_1|. \qquad\qquad\qquad \text{Fact 4}
\end{aligned}$$

$\square$

Lemma 2 shows that rejecting $\mathcal{R}$ is sufficient to produce rejection sets that have $\mathrm{TPR}(\mathcal{S}) \geq 1-\delta$. Thus, our goal in this proof is to show a bound on $T := \min\{t \in \mathbb{N} : \mathcal{R} \subseteq \mathcal{S}_t\}$ with at least $1-\delta$ probability, where $T = \infty$ if $\mathcal{R} \not\subseteq \mathcal{S}_t$ for all $t \in \mathbb{N}$. Note that for all $t \geq T$, $\mathcal{S}_T \subseteq \mathcal{S}_t$ by the way (5a) is defined — it does not sample arms that have already been rejected.

We note that we can use any $\varphi$ defined in Fact 4 in this proof and still achieve the desired result. For simplicity, we use $\varphi$ to denote $\varphi^0$ in this proof. First we define a notion of inverse for $\varphi$. Let

$$\varphi^{-1}(\epsilon,\delta) := \min\{t : \varphi(t,\delta) \leq \epsilon\}. \tag{9}$$

JJ and other work [18] show that for some absolute constant $c > 0$,

$$\varphi^{-1}(\epsilon,\delta) \leq c\epsilon^{-2}\log(\log(\epsilon^{-2})/\delta) \text{ for all } \epsilon \in \mathbb{R}^+, \delta \in (0,1). \tag{10}$$

Also, recall that $f \lesssim g$ denotes $f$ asymptotically dominates $g$ i.e. there exist $c > 0$ that is independent of the problem parameters such that $f \leq cg$.

We decompose $T$ into the number of time steps the algorithm samples a null arm, and the number of time steps the algorithm samples a non-null arm:

$$T = \sum_{t=1}^{\infty} \mathbf{I}\{\mathcal{R} \not\subseteq \mathcal{S}_t\} = \sum_{t=1}^{\infty} \mathbf{I}\{I_t \in \mathcal{H}_0, \mathcal{R} \not\subseteq \mathcal{S}_t\} + \mathbf{I}\{I_t \in \mathcal{H}_1, \mathcal{R} \not\subseteq \mathcal{S}_t\}. \tag{11}$$

Our first goal is to prove a sample complexity bound on $\sum_{t=1}^{\infty} \mathbf{I}\{I_t \in \mathcal{H}_0, \mathcal{R} \not\subseteq \mathcal{S}_t\}$. We define the following variables for each $i \in [k]$.

$$\rho_i := \inf\{\rho \in [0,1] : |\widehat{\mu}_{i,t} - \mu_i| > \varphi(t,\rho) \text{ for all } t \in \mathbb{N}\} \cup \{1\}. \tag{12}$$

**Lemma 3.** *For each $i \in [k]$, $\mathbb{P}(\rho_i \leq s) \leq s$ for $s \in (0,1)$ i.e. $\rho_i$ is superuniformly distributed.*

The above lemma follows directly from Fact 4. We also define a concentration bound for independent superuniformly distributed variables.

**Lemma 4.** *For any fixed positive reals $a_1,\ldots,a_d$, independent superuniformly distributed random variables $r_1,\ldots,r_d$, and $\beta \in (0,1)$, the following event occurs with probability at least $1-\beta$:*

$$\sum_{i=1}^{d} a_i\log(1/r_i) \leq 5\log(1/\beta)\sum_{i=1}^{d} a_i.$$

This lemma follows directly from recognizing $a_i\log(1/r_i)$ satisfies the requirements for $Z_i$ in Lemma 1. Now, we will show that the UCB for each $i \in \mathcal{R}$ will be above $\mu_i$.

**Lemma 5.** *Let $\nu_i$ be sub-Gaussian for each $i \in [k]$. Any algorithm with $(\mathcal{A}_t)$ that outputs $\mathcal{I}_t = \{I_t\}$ as defined in (5a) has the following property:*

$$\sum_{t=1}^{\infty} \mathbf{I}\{I_t \in \mathcal{H}_0, \mathcal{R} \nsubseteq \mathcal{S}_t\} \lesssim \sum_{i \in \mathcal{H}_0} \Delta_i^{-2} \log(\log(\Delta_i^{-2})/\delta\rho_i).$$

*Proof.* The following is true for any $i \in \mathcal{R}$ and $t \in \mathbb{N}$:

$$\begin{aligned}
\widehat{\mu}_{i,t} + \varphi(T_i(t),\delta)) &\geq \mu_i - \varphi(T_i(t),\rho_i) + \varphi(T_i(t),\delta)) && \text{by def. of } \rho_i \text{ and } \mathcal{R} \\
&\geq \mu_i. && \text{by def. of } \mathcal{R}
\end{aligned}$$

Thus, $\{\mathcal{R} \nsubseteq \mathcal{S}_t\}$ implies for any $t \in \mathbb{N}$:

$$\begin{aligned}
\text{argmax}_{i \in [k] \backslash \mathcal{S}_t} \widehat{\mu}_{i,t} + \varphi(T_i(t),\delta) &\overset{(i)}{\geq} \min_{i \in \mathcal{R}} \mu_i \\
&\geq \min_{i \in \mathcal{H}_1} \mu_i, && (13)
\end{aligned}$$

where inequality (i) is by the definition of $\mathcal{R}$.

In addition, we argue that the UCB for $i \in \mathcal{H}_0$ will shrink below $\min_{i \in \mathcal{H}_1} \mu_i$ quickly. For $i \in \mathcal{H}_0$, the following is true for any $t \in \mathbb{N}$:

$$\begin{aligned}
\widehat{\mu}_{i,t} + \varphi(T_i(t),\delta) &\leq \mu_i + \varphi(T_i(t),\rho_i) + \varphi(T_i(t),\delta) \\
&\leq \mu_i + 2\varphi(T_i(t),\delta\rho_i). && (14)
\end{aligned}$$

Thus, $\{\forall i \in \mathcal{H}_0 : \mu_i + 2\varphi(T_i(t),\delta\rho_i) \leq \min_{i \in \mathcal{H}_1} \mu_i, \mathcal{R} \nsubseteq \mathcal{S}_t\} \implies \{I_t \in \mathcal{H}_1\}$ for all $t \in \mathbb{N}$ by (13) and (14).

Subsequently, we argue the following:

$$\begin{aligned}
\sum_{t=1}^{\infty} \mathbf{I}\{I_t \in \mathcal{H}_0, \mathcal{R} \nsubseteq \mathcal{S}_t\} &\leq \sum_{t=1}^{\infty} \mathbf{I}\left\{\exists i \in \mathcal{H}_0 : \mu_i + 2\varphi(T_i(t),\delta\rho_i) > \min_{i \in \mathcal{H}_1} \mu_i, \mathcal{R} \nsubseteq \mathcal{S}_t\right\} \\
&\leq \sum_{i \in \mathcal{H}_0} \varphi^{-1}(\Delta_i/2,\delta\rho_i) && \mu_i \leq \mu_0 \text{ for all } i \in \mathcal{H}_0 \\
&\lesssim \sum_{i \in \mathcal{H}_0} \Delta_i^{-2} \log(\log(\Delta_i^{-2})/\delta\rho_i).
\end{aligned}$$

Thus, we have shown our desired result. $\qquad\square$

Now, we proceed to show a bound on $\sum_{t=1}^{\infty} \mathbf{I}\{I_t \in \mathcal{H}_1, \mathcal{R} \nsubseteq \mathcal{S}_t\}$. Denote $\pi$ as an arbitrary mapping from $\mathcal{H}_1$ to $[|\mathcal{H}_1|]$. Let $(x)_+ = x \vee 0$ for any $x \in \mathbb{R}$. We define additional variables as follows:

$$\ell_i' := (\lceil \log(2\Delta_i^{-1}) - 5/2 \rceil)_+,$$

$$\rho_i^{\text{DM}} := \min_{t \in \mathbb{N}} \frac{1}{\exp\left(\sum_{j=1}^{T_i(t)} \lambda_{\ell_i'}(\mu_i - X_{i,t_i(j)}) - \lambda_{\ell_i'}^2/2\right)}.$$

**Lemma 6.** $\mathbb{P}(\rho_i^{\text{DM}} \leq s) \leq s$ *for $s \in (0,1)$ i.e. $\rho_i^{\text{DM}}$ is superuniformly distributed for each $i \in \mathcal{H}_1$.*

*Proof.* First, we prove an underlying process is a nonnegative supermartingale. Let

$$M_t = \exp\left(\sum_{j=1}^{T_i(t)} \lambda_{\ell_i'}(\mu_i - X_{i,t_i(j)}) - \lambda_{\ell_i'}^2/2\right).$$

Assume that arm $i$ is selected at time $t$ — otherwise the supermartingale property is directly satisfied.

$$\mathbb{E}[M_t \mid \mathcal{F}_{t-1}] = \mathbb{E}\left[\exp\left(\sum_{j=1}^{T_i(t)} \lambda_{\ell_i'}(\mu_i - X_{i,t_i(j)}) - \lambda_{\ell_i'}^2/2\right) \mid \mathcal{F}_{t-1}\right]$$

$$= \mathbb{E}\left[\exp\left(\lambda_{\ell_i'}(\mu_i - X_{i,t}) - \lambda_{\ell_i'}^2/2\right) \mid \mathcal{F}_{t-1}\right]\exp\left(\sum_{j=1}^{T_i(t-1)} \lambda_{\ell_i'}(\mu_i - X_{i,t_i(j)}) - \lambda_{\ell_i'}^2/2\right)$$

$$\leq M_{t-1},$$

where the final inequality holds because $X_{i,t}$ are i.i.d. across $t \in \mathbb{N}$, have mean $\mu_i$, and are 1-sub-Gaussian.

Thus, $\rho_i^{\mathrm{DM}}$ is a superuniform random variable by applying Ville's inequality to $(M_t)$.

$\square$

**Proposition 8** (Growth of $E_{i,t}^{\mathrm{DM}}$). *When $i \in \mathcal{H}_1$ and $\nu_i$ is 1-sub-Gaussian,*

$$\log E_{i,t} \gtrsim \Delta_i^2 T_i(t) - \log\log(\Delta_i^{-2}) - \log(1/\rho_i^{\mathrm{DM}}).$$

*Proof.* We show the following lower bound on $E_{i,t}^{\mathrm{DM}}$:

$$E_{i,t}^{\mathrm{DM}} = \sum_{\ell=0}^{\infty} w_\ell \exp(T_i(t)(\lambda_\ell \Delta_i - \lambda_\ell^2))\exp\left(\sum_{j=1}^{T_i(t)} \lambda_\ell^2/2 - \lambda_\ell(\mu_i - X_{i,t_i(j)})\right)$$

$$\geq w_{\ell_i'}\exp(T_i(t)(\lambda_{\ell_i'}\Delta_i - \lambda_{\ell_i'}^2))\exp\left(\sum_{j=1}^{T_i(t)} \lambda_{\ell_i'}^2/2 - \lambda_{\ell_i'}(\mu_i - X_{i,t_i(j)})\right)$$

$$\geq \exp\left(\frac{1}{4e}\Delta_i^2 T_i(t) - \log(1/w_{\ell_i'}) - \log(1/\rho_i^{\mathrm{DM}})\right). \qquad \text{by def. of } \ell_i' \text{ and } \rho_i^{\mathrm{DM}}$$

Thus, plugging in $w_{\ell_i'}$, we get our desired result. $\square$

*Proof of Theorem 1.* By Proposition 8,

$$T_i(t) \gtrsim \Delta_i^{-2}\log(\varepsilon\log(\Delta_i^{-2})/\rho_i^{\mathrm{DM}})$$

implies $E_{i,t} \geq \varepsilon$ for $\varepsilon > 0$.

Now, we can derive the following bound:

$$\sum_{t=1}^{\infty} \mathbf{I}\{\mathcal{R} \not\subseteq \mathcal{S}_t\} = \sum_{t=1}^{\infty} \mathbf{I}\{I_t \in \mathcal{H}_0, \mathcal{R} \not\subseteq \mathcal{S}_t\} + \sum_{t=1}^{\infty} \mathbf{I}\{I_t \in \mathcal{H}_1, \mathcal{R} \not\subseteq \mathcal{S}_t\}$$

$$\lesssim \sum_{i \in \mathcal{H}_0} \Delta_i^{-2}\log\log(\Delta_i^{-2}) + \Delta_i^{-2}\log(1/\rho_i) + \Delta_i^{-2}\log(1/\delta)$$

$$+ \max_{\pi} \sum_{i \in \mathcal{H}_1} \Delta_i^{-2}\log\log(\Delta_i^{-2}) + \Delta_i^{-2}\log(1/\rho_i^{\mathrm{DM}}) + \Delta_i^{-2}\log(k/\pi(i)\delta).$$

by Lemma 5

Recall that $\rho_i$, by Lemma 3, and $\rho_i^{\mathrm{DM}}$, by Lemma 6, are superuniform random variables that are independent across $i \in [k]$ and $i \in \mathcal{H}_1$, respectively. Consequently, we can apply Lemma 4 at level $\beta = \delta/2$ to $\rho_i$ for $i \in [k]$ and $\rho_i^{\mathrm{DM}}$ for $i \in \mathcal{H}_1$. Then, the following happens with at least $1 - \delta$ probability:

$$\sum_{t=1}^{\infty} \mathbf{I}\{\mathcal{R} \not\subseteq \mathcal{S}_t\} \lesssim \sum_{i \in \mathcal{H}_0} \Delta_i^{-2}\log(\log(\Delta_i^{-2})/\delta)$$

$$+ \max_{\pi} \sum_{i \in \mathcal{H}_1} \Delta_i^{-2}\log(\log(\Delta_i^{-2})/\delta) + \Delta_i^{-2}\log(k/\pi(i)).$$

**Algorithm 2:** An generic algorithm that uses a BAI subroutine to find a best arm that is not in the rejection set for the algorithm to then repeatedly sample and eventually reject.

---

**Input:** A BAI algorithm $\mathcal{B}$ that takes in $B \subseteq [k]$, and a history of samples and initial randomness $D_t(B) := U \cup \{(i,j,X_{i,j}) : j \leq t, i \in \mathcal{I}_j \cap B\}$. At each step, $\mathcal{B}$ outputs a superarm $\mathcal{I} \in \mathcal{K}$ to sample next, or a best arm $i \in B$. Let $\delta \in (0,1)$ be the level of FDR control and $\delta' \in (0,1)$ be the corrected level for p-variables. Let $(e_{i,t})$ and $(p_{i,t})$ be realized values of e-processes and p-processes, respectively, for each $i \in [k]$. Let $\tau^* \in \mathcal{T}$ be the stopping time for the algorithm.

**Initialize** $\mathcal{S}_0 := \emptyset$
**Initialize** bestarm $:=$ **none**
**for** $t \in 1,...,$ **do**

    $B := [k] \setminus \mathcal{S}_{t-1}$
    **if** bestarm *is **none** or* bestarm $\in \mathcal{S}_{t-1}$ **then**
        $\mathcal{I}_t := \mathcal{B}(B, D_{t-1}(B))$
        **if** $\mathcal{B}(B, D_{t-1}(B))$ *terminated with best arm* $I_t$ **then** bestarm $:= I_t$;
    **else**
        $\mathcal{I}_t := \{\text{bestarm}\}$ (or an arbitrary $\mathcal{I} \in \mathcal{K}$ such that bestarm $\in \mathcal{K}$).
    **end**
    Update e-process or p-process for each queried arm not in $\mathcal{S}_{t-1}$.
    $\mathcal{S}_t := \begin{cases} \text{BH}[\delta'](p_{1,t},...,p_{k,t}) \text{ or arbitrary p-self-consistent set} & \text{if using p-variables} \\ \text{eBH}[\delta](e_{1,t},...,e_{k,t}) \text{ or arbitrary e-self-consistent set} & \text{if using e-variables} \end{cases}$
    **if** $\tau^* = t$ **then** stop and **return** $\mathcal{S}_t$;
**end**

---

We can derive two different bounds. The first is using the fact that $\sum_{i=1}^{|\mathcal{H}_1|} \log(k/i) \leq k$. As a result,

$$\sum_{t=1}^{\infty} \mathbf{I}\{\mathcal{R} \not\subseteq \mathcal{S}_t\} \lesssim k\Delta^{-2}\log(\log(\Delta^{-2})/\delta).$$

The second comes from dropping the $\pi(i)$ term, which is as follows:

$$\sum_{t=1}^{\infty} \mathbf{I}\{\mathcal{R} \not\subseteq \mathcal{S}_t\} \lesssim \sum_{i \in \mathcal{H}_0} \Delta_i^{-2}\log(\log(\Delta_i^{-2})/\delta) + \sum_{i \in \mathcal{H}_1} \Delta_i^{-2}\log(k\log(\Delta_i^{-2})/\delta).$$

Thus, we have shown both sample complexity bounds as desired. $\qquad \square$

## C  Generic algorithms for $(\mathcal{A}_t)$

We propose two generic algorithms that can be used for the exploration component in Algorithm 1 regardless of the type of hypotheses tested or what the joint distribution of $X_{1,t},...,X_{k,t}$ is. For simplicity, we assume that the algorithm can always sample each arm separately, i.e. $\{\{1\},...,\{k\}\} \subseteq \mathcal{K}$.

**Reduction to best arm identification (BAI)**  The first relies on having access to a best arm identification (BAI) algorithm. BAI is well studied problem, and there exist many algorithms for it in both the standard bandit setting [1, 22, 18, 23] and combinatorial bandit settings [12, 10, 20]. A BAI algorithm returns the "best arm" i.e. the arm with the highest mean reward, with high probability. Thus, we can employ a BAI algorithm as a subroutine to repeatedly find the best arm out of arms not in the rejection set, and then repeatedly sample that best arm until it is rejected. Algorithm 2 formulates an algorithm using a BAI subroutine that fits the meta-algorithm introduced in Algorithm 1 . Consequently, we can immediately have access to algorithms for multiple testing that have non-trivial exploration components for a wide variety of settings.

**Largest e-process (or smallest p-process)**  If no apparent exploration strategy exists, we can always select the arm that currently has the most evidence for rejection, but has not yet been rejected. Algorithm 3 illustrates this algorithm — our exploration strategy is to simply pick the superarms that

---

**Algorithm 3:** An generic algorithm applicable to any combinatorial bandit and set of hypotheses that utilizes the evidence itself (i.e. p-variables or e-variables) to select arms to sample.

---

**Input:** Let $\delta \in (0,1)$ be the level of FDR control and $\delta' \in (0,1)$ be the corrected level for p-variables. $(e_{i,t})$ and $(p_{i,t})$ be realized values of e-processes and p-processes, respectively, for each $i \in [k]$. Let $\tau^* \in \mathcal{T}$ be the stopping time for the algorithm.

**Initialize** $\mathcal{S}_0 := \emptyset$
**Initialize** $e_{i,0} = 1$ or $p_{i,0} = 1$ for all $i \in [k]$
**for** $t \in 1,...,$ **do**

    **if** $t \leq k$ **then**

        $I_t := t$
        $\mathcal{I}_t := \{I_t\}$ or an arbitrary $\mathcal{I} \in \mathcal{K}$ where $I_t \in \mathcal{I}$

    **else**

        $I_t \in \begin{cases} \operatorname{argmin}_{i \in [k] \setminus \mathcal{S}_{t-1}} p_{i,t-1} & \text{if using p-variables} \\ \operatorname{argmax}_{i \in [k] \setminus \mathcal{S}_{t-1}} e_{i,t-1} & \text{if using e-variables} \end{cases}$ (an arbitrary element of
        argmin/argmax).
        $\mathcal{I}_t := \{I_t\}$ or an arbitrary $\mathcal{I} \in \mathcal{K}$ where $I_t \in \mathcal{I}$

    **end**

    Update e-process or p-process for each queried arm not in $\mathcal{S}_{t-1}$.

    $\mathcal{S}_t := \begin{cases} \text{BH}[\delta'](p_{1,t},...,p_{k,t}) \text{ or arbitrary p-self-consistent set} & \text{if using p-variables} \\ \text{eBH}[\delta](e_{1,t},...,e_{k,t}) \text{ or arbitrary e-self-consistent set} & \text{if using e-variables} \end{cases}$

    **if** $\tau^* = t$ **then** stop and **return** $\mathcal{S}_t$;

**end**

---

contains the arm that already has the "most" evidence (largest e-value or smallest p-value). Thus, simply having e-variables or p-variables for the hypotheses we are testing can be used to inform the sampling strategy.

Both of the aforementioned algorithms guarantee FDR control due to being instances of Algorithm 1.

**Proposition 9.** *Algorithms 2 and 3 guarantee that* $\sup_{\tau \in \mathcal{T}} \text{FDR}(\mathcal{S}_\tau) \leq \delta$.

As a result, we always have a default choice of exploration component if we are unaware of any domain specific strategies for sampling.

## D   Extensions on the bandit setting

In this section, we consider some special cases and extensions on the bandit settings. This includes settings involving streaming data, constrained rejection sets, multiple agents, and hypotheses involving multiples arms. Critically, we show how our framework can be easily adaptable to each of these settings to still maintain valid FDR guarantees.

### D.1   Streaming data setting

A unique instance of the combinatorial bandits is the streaming data setting, where the algorithm has access to the all rewards at each time step. Instead of choosing a sampling policy, the algorithm can choose a stopping time $\tau_i$ for each arm $i \in [k]$ that marks when the algorithm will cease observing arm $i$. Although $X_{1,t},...,X_{k,t}$ may be arbitrarily dependent, Algorithm 1 with e-variables can still use all the observations from each arm at each time step. This is because e-BH on e-variables maintains the same FDR control irrespective of dependence structure. Thus, we can propose a simple strategy in Algorithm 4 that stops the monitoring of an arm once that arm has been rejected by e-BH, and can limit the amount of time between rejections or total time run before the algorithm stops. By Proposition 4, we have the following FDR guarantee.

**Proposition 10.** *Algorithm 4 ensures* $\sup_{\tau \in \mathcal{T}} \text{FDR}(\mathcal{S}_\tau^*) \leq \delta$.

Bartroff and Song [5] also study multiple testing in the streaming data setting, and prove FDR guarantees similar to Proposition 10 for an algorithm that is virtually identical to Algorithm 4 with p-variables and BH. A key difference between their results and ours is that they use test statistics in their algorithm instead of p-variables, and make assumptions about the power of the test statistics that also allow them to provide guarantees about the false negative rate i.e. the expected proportion

of hypotheses that are not rejected which are true discoveries. Thus, our framework for FDR control subsumes existing methods for the streaming setting. Other error metrics such as family-wise error rate and probabilistic bounds on the FDP have also been studied in the sequential setting [3, 4, 2].

---

**Algorithm 4:** An algorithm for monitoring in the streaming data setting. This algorithm stops when the maximum time $t_{\max}$ has been reached, or more than $t_{\mathrm{gap}}$ steps have passed since the last rejection. Once an arm is added to $\mathcal{S}_t$, the algorithm stops monitoring it.

---

$\mathcal{S}_0 = \emptyset$
$t_{\mathrm{prev.rejection}} = 0$
**for** $t \in 1, ..., t_{\max}$ **do**
$\quad \mathcal{I}_t := [k] \setminus \mathcal{S}_{t-1}$
$\quad \mathcal{S}_t := \begin{cases} \mathrm{eBH}[\delta](e_{1,t}, ..., e_{k,t}) & \text{if using e-variables} \\ \mathrm{BH}[\delta'](p_{1,t}, ..., p_{k,t}) & \text{if using p-variables} \end{cases}$
$\quad$ **if** $t - t_{\mathrm{prev.rejection}} > t_{\mathrm{gap}}$ **or** $\mathcal{S}_t = [k]$ **then return** $\mathcal{S}_t$;
**end**
**return** $\mathcal{S}_t$

---

### D.2  Structured rejection sets

Structured rejection sets arise in problems where there is a fixed hierarchy that restrict the sets of hypotheses that can be rejected e.g. hypothesis 2 can only be rejected if hypothesis 1 is rejected also. Recent work in multiple testing with FDR control has studied settings with general structural constraints [25] and when the constraints have been restricted to form a directed acyclic graph (DAG) [30]. A DAG constraint requires all predecessors of a hypothesis in the DAG to be rejected before the hypothesis itself can be rejected. Thus, the algorithm does not necessarily output the result of BH or e-BH, but rather a p-self-consistent or e-self-consistent set, respectively. Table 2 illustrates the FDR guarantees for p-variables in the structured setting for different dependence relationsips between $X_{1,t}, ..., X_{k,t}$. In case with adaptive $(\mathcal{A}_t)$ and $\tau^*$ when $X_{1,t}...,X_{k,t}$ are independent — the guarantee in that setting remains unchanged due to the proof of FDR control already being based upon the fact that the output of BH was p-self-consistent to $P_1^*, ..., P_k^*$. Similarly, e-variables still do not pay a penalty when moving from e-BH to an arbitrary e-self-consistent set. The FDR when using e-variables remains below $\delta$ after setting $\alpha = \delta$.

Table 2: FDR control guaranteed by an arbitrary p-self-consistent set, and the $\delta'$ to ensure $\delta$ control of FDR in Algorithm 1 under different dependence structures and adaptivity of $(\mathcal{A}_t)$. Adaptive and non-adaptive strategies no longer have different guarantees when outputting a p-self-consistent set. On the other hand, the FDR control of an e-self-consistent set remains unchanged at $\alpha = \delta$.

| **Adaptivity of** $(\mathcal{A}_t)$ **and** $\tau^*$ | **Dependence of** $X_{1,t}, ..., X_{k,t}$ | |
| --- | --- | --- |
| | *independent* | *arbitrarily dependent* |
| *adaptive* | $\mathrm{FDR}(\mathcal{S}) \le \alpha((1 + \log(1/\alpha)) \wedge \log k)$ | $\mathrm{FDR}(\mathcal{S}) \le \alpha \log k$ |
| *non-adpative* | $\delta' = c_\delta \vee (\delta/\log k)$ (Prop. 1) | $\delta' = c_\delta / \log k$ |

We show an example in Figure 3 of a set of hypotheses in a DAG structure. Thus, a hypothesis can only be rejected if its predecessors in the DAG are also rejected. We compare the output of BH, e-BH, and both the largest e-self-consistent set and p-self-consistent set that respect the DAG constraints. The e-values are calculated assuming that the p-variables are reciprocals of e-variables. We assume the p-variables are all arbitrarily dependent. The largest e-self-consistent set and the largest p-self-consistent set are simply the largest subset of e-BH and BH, respectively, that satisfies the DAG constraints.

### D.3  Multiple agents

There are many scenarios where multiple agents are interacting with the same bandit and we hope to have the agents cooperatively accumulate evidence. For example, a research group could be interested in resuming the study of a hypothesis that previous researchers have run experiments on, and would like to combine existing evidence with the new evidence they collect from their own experiments. A cooperative situation could also arise when there are multiple groups that each work on a subset of some overarching set of hypotheses — the groups can combine the evidence they have for each hypothesis. In these cases, the evidence shared, either from previous studies or concurrent collaborators, might only

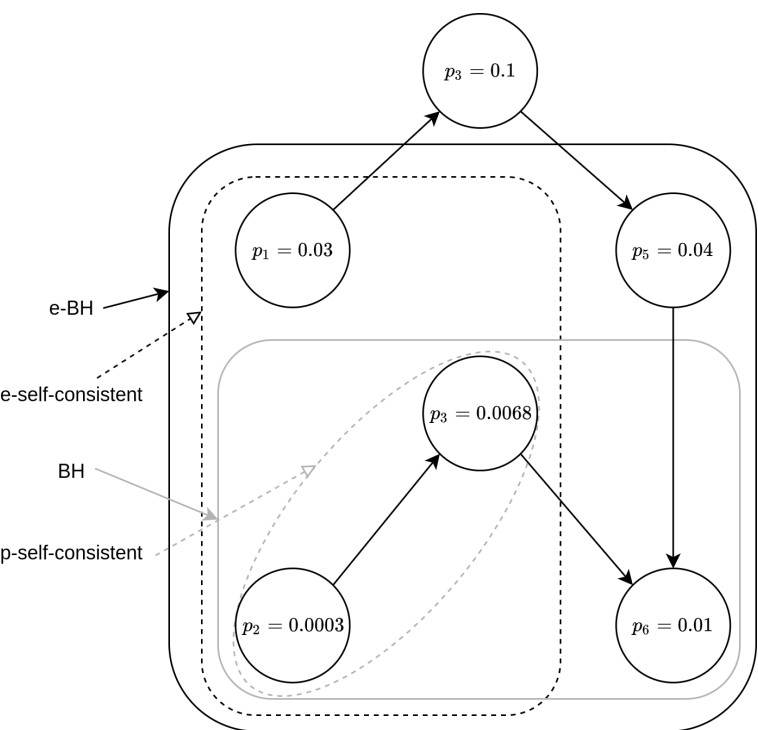

Figure 3: Example set of p-values for hypotheses that have a DAG constraint upon them and rejection sets that ensure $\mathrm{FDR}(\mathcal{S}) \leq \delta = 0.05$. We assume the p-variables are arbitrarily dependent, and are reciprocals of e-variables for the sake of comparing e-variable vs. p-variable procedures. The e-self-consistent and p-self-consistent rejection sets are the largest such sets that satisfy the $\mathrm{FDR}$ guarantee and the DAG constraints. The e-BH and BH rejection sets violate the DAG constraints, i.e. they are not valid rejection sets, but they do maintain $\mathrm{FDR}(\mathcal{S}) \leq \delta$. The largest valid e-self-consistent rejection set and p-self-consistent rejection set are simply the largest subsets that satisfy the DAG constraints of e-BH and BH, respectively.

be in the form of an e-value or p-value — the actual samples may be obfuscated for privacy reasons. Thus, each of the scenarios require the merging of multiple statistics (from each agent) into a single statistic representing the total amount of evidence for rejecting a hypothesis.

Assume we have $m$ agents and let $E_1,...,E_m$ denote the e-variables all testing the same hypothesis. If the e-variables are all independent, we can define an ie-merging function (outputs an e-variable from independent e-variables) $f_{\mathrm{prod}}$ as follows:

$$f_{\mathrm{prod}}(E_1,...,E_m) := \prod_{i=1}^{m} E_i.$$

**Proposition 11.** *If $E_1,...,E_m$ are independent e-variables, then $f_{\mathrm{prod}}(E_1,...,E_m)$ is also an e-variable.*

The above proposition follows from the fact that the expectation of the product is the product of expectation for independent random variables.

If $E_1,...E_m$ are dependent, then we can define the following e-merging function (outputs an e-variable from *arbitrarily dependent* e-variables):

$$f_{\mathrm{mean}}(E_1,...,E_m) := \frac{1}{m}\sum_{i=1}^{m} E_m.$$

**Proposition 12.** *If $E_1,...,E_m$ are arbitrarily dependent e-variables, then $f_{\mathrm{mean}}(E_1,...,E_m)$ is also an e-variable.*

Vovk and Wang [41] show that the set of functions corresponding to all convex combinations of $f_{\mathrm{mean}}$ and 1 are the only admissible e-merging functions in the class of all symmetric e-merging functions.

They also show a weaker sense of dominance for $f_{\text{prod}}$ — they prove it outputs a larger e-value than any other symmetric ie-merging function if all the input e-values are at least 1. Thus, e-variables can be merged in a relatively simple fashion without many assumptions.

On the other hand, merging p-variables is difficult. When the p-variables are independent, Birnbaum [8] show that any valid merging function which is monotonic w.r.t. to the p-values is admissible. When the p-variables are arbitrarily dependent, Vovk et al. [43] prove that there are also many admissible symmetric p-merging functions. Consequently, the p-variable picture is much less clear about how to optimally merge p-variables, particularly when there is arbitrary dependence among them. To illustrate these e-merging/ie-merging functions can be used in a bandit setting, we consider an example multi-agent problem where many research groups are submitting studies to the same journal.

### D.3.1  Example: controlling the FDR of results in a journal

We consider a situation where the editors of a journal are interested in guaranteeing the accuracy of the results published within the journal. Specifically, they aim to ensure FDR control on the discoveries within the papers accepted to the journal. The journal requires that each study that is submitted is also accompanied by an e-value. Since many groups can be testing the same hypothesis, the journal can use the aforementioned merging techniques to combine the e-values reported by different groups and produce a valid, aggregate e-variable.

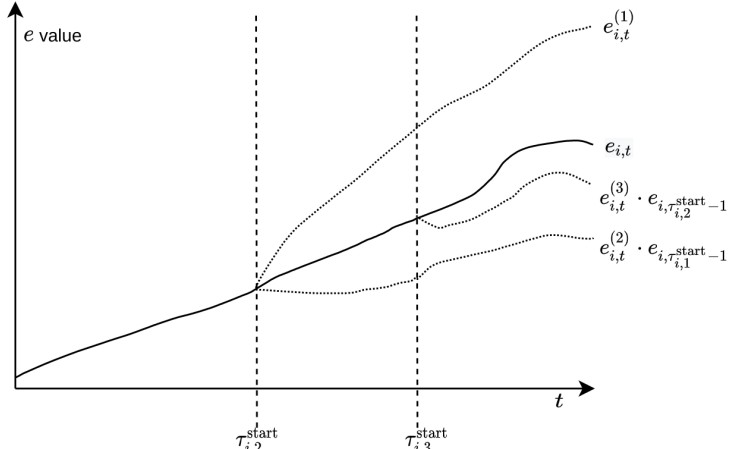

Figure 4: An illustration of how $e_{i,t}$ changes in relation to each $e_{i,t}^{(\ell)}$ for a case where $\ell = 3$ in Algorithm 5. We see that $e_{i,t}$ and $e_{i,t}^{(1)}$ are identical up to $\tau_{i,2}^{\text{start}} - 1$, where agent 2 begins to sample arm $i$. Agent 2's process starts at $e_{i,\tau_{i,2}^{\text{start}}-1}$. Similarly, Agent 3's process starts at $e_{i,\tau_{i,3}^{\text{start}}-1}$ when it starts to sample $i$ as well. Validity of Algorithm 5 arises from the fact that each new agent $\ell$ has its own $e_{i,t}^{(\ell)}$ scaled by the $e_{i,t}$ that has been achieved already.

**Formalizing the multi-agent setup**  The reward of the $i$th arm on the $t$th day for the $\ell$th agent is denoted as $X_{i,t}^{(\ell)}$ for all $t, \ell \in \mathbb{N}$ and $i \in [k]$. We let the index for agents, $\ell$, be in $\mathbb{N}$ to allow for arbitrarily large, but finite, number of agents at each time step. We let the joint distribution of $X_{1,t}^{(\ell)}, ..., X_{k,t}^{(\ell)}$ be identically distributed across $\ell, t \in \mathbb{N}$. Consequently, the rewards $X_{1,t}^{(\ell)}, ..., X_{k,t}^{(\ell)}$ corresponding to each agent $\ell$ are identical in a marginal sense across all $\ell \in \mathbb{N}$. However, there can be arbitrary dependencies between the rewards of different agents. Thus, we allow for a setting where, for each $i \in [k]$ and $t \in \mathbb{N}$, $X_{i,t}^{(\ell)}$ is the same reward across all $\ell \in \mathbb{N}$, and a setting where $X_{i,t}^{(\ell)}$ are independent across $\ell \in \mathbb{N}$. Each agent $\ell \in \mathbb{N}$ outputs $\mathcal{I}_t^{(\ell)} \in \mathcal{K} \cup \{\emptyset\}$ for each $t \in \mathbb{N}$. Let the set of agents (e.g. set of studies) on day $t$ that are testing hypothesis $i$ be $A_{i,t}$ for each $t \in \mathbb{N}$ and $i \in [k]$. Critically, we require that $A_{i,t}$ be of finite cardinality almost surely and predictable w.r.t. the new canonical filtration $(\mathcal{G}_t)$. We define the canonical filtration for the multi-agent setting as follows:

$$\mathcal{G}_t := \sigma(U \cup \{(i,s,\ell,X_{i,s}^{(\ell)}) : i \in \mathcal{I}_s^{(\ell)}, s \le t, \ell \in A_{i,s}\}).$$

We denote the e-process of the $\ell$th agent for hypothesis $i$ to be $(E_{i,t}^{(\ell)})$, where $E_{i,t}^{(\ell)} = 1$ if $\ell \notin A_{i,t}$. Implicitly, there exists a stopping time $\tau_{i,\ell}^{\mathrm{start}}$ w.r.t. $(\mathcal{G}_t)$ that denotes the time when the $\ell$th agent begins testing the $i$th hypothesis for $\ell \in \mathbb{N}$ and $i \in [k]$ (i.e. the time of the first sample of arm $i$ by agent $\ell$). Algorithm 5 explicitly formulates the algorithm for dealing with e-values coming from multiple agents.

---

**Algorithm 5:** An algorithm for aggregating evidence in the form of e-values from many agents. The algorithm takes the mean of the e-values for each hypothesis on each day and applies an e-self-consistent procedure to these aggregated e-values to maintain valid FDR control at $\delta$.

---

**Input:** A level of control $\delta$ in $(0,1)$. $(e_{i,t}^{(\ell)})$ are the realized values of an e-process for $\ell \in \mathbb{N}, i \in [k]$.
**Initialize** $e_{i,0} := 1$ for $i \in [k]$
$\mathcal{S}_0 := \emptyset$
**for** $t \in 1,\dots$ **do**

> Receive new results from new or existing agents and update $e_{i,t}^{(j)}$ for $i \in [k]$ and $j \in [a_t]$
> **for** $i \in [k]$ **do**
>
> > $e_{i,t} := \begin{cases} \frac{1}{|A_{i,t}|} \sum\limits_{j \in A_{i,t}} e_{i,\tau_{i,j}^{\mathrm{start}}-1} \cdot e_{i,t}^{(\ell)} & \text{if } i \notin \mathcal{S}_{t-1} \\ e_{i,t-1} & \text{else} \end{cases}$.
>
> **end**
> $\mathcal{S}_t := \mathrm{eBH}[\delta](e_{1,t},\dots,e_{k,t})$ or an arbitrary e-self-consistent set.

**end**

---

Figure 4 illustrates how $e_{i,t}$ behaves w.r.t. to the $e_{i,t}^{(\ell)}$ of each agent in Algorithm 5. Algorithm 5 uses a merging approach that is in between $f_{\mathrm{prod}}$ and $f_{\mathrm{mean}}$. Intuitively, we know the rewards across $t \in \mathbb{N}$ are independent, and consequently we can merge e-values by taking the product. When merging across different $\ell \in \mathbb{N}, i \in [k]$, however, there may be arbitrary dependence between rewards. Consequently, we must take the mean of those e-values. From a betting perspective as discussed in Shafer [33], we can view our algorithm as splitting the current wealth (current $e_{i,t}$) evenly across each agent whenever a new agent is introduced before allowing each agent to continue or begin its own strategy. Regardless, we can show the following guarantee concerning Algorithm 5.

**Proposition 13.** *Let* $(E_{i,t}^{(\ell)})$ *be upper bounded by some nonnegative supermartingale* $(M_{i,t}^{(\ell)})$ *w.r.t.* $(\mathcal{G}_t)$ *for* $i \in [k], \ell \in \mathbb{N}$ *where* $E_{i,t}^{(\ell)} = M_{i,t}^{(\ell)} = 1$ *for* $t < \tau_{i,\ell}^{\mathrm{start}}$. *Algorithm 5 ensures that* $\sup_{\tau \in \mathcal{T}} \mathrm{FDR}(\mathcal{S}_\tau) \leq \delta$.

*Proof.* Define $M_{i,t} := \frac{1}{|A_{i,t}|} \sum_{\ell \in A_{i,t}} M_{i,t}^{(\ell)} \cdot M_{i,\tau_{i,\ell}^{\mathrm{start}}-1}$ when $i \notin \mathcal{S}_{t-1}$ and $M_{i,t} := M_{i,t-1}$ otherwise. We can see that $M_{i,t}$ upper bounds $E_{i,t}$ for all $t \in \mathbb{N}, i \in [k]$. We will show that $(M_{i,t})$ is a nonnegative supermartingale. Assume that we have not rejected the $i$th hypothesis yet, since otherwise $M_{i,t} = M_{i,t-1}$, which satisfies the supermartingale property.

$$
\begin{aligned}
\mathbb{E}[M_{i,t} \mid \mathcal{G}_{t-1}] &= \mathbb{E}\left[ \frac{1}{|A_{i,t}|} \sum_{\ell \in A_{i,t}} M_{i,t}^{(\ell)} \cdot M_{i,\tau_{i,\ell}^{\mathrm{start}}-1} \,\Big|\, \mathcal{G}_{t-1} \right] \\
&= \frac{1}{|A_{i,t}|} \left( \sum_{\ell \in A_{i,t-1}} \mathbb{E}\left[ M_{i,t}^{(\ell)} \cdot M_{i,\tau_{i,\ell}^{\mathrm{start}}-1} \mid \mathcal{G}_{t-1} \right] + \sum_{\ell \in A_{i,t} \setminus A_{i,t-1}} \mathbb{E}\left[ M_{i,t}^{(\ell)} \cdot M_{i,t-1} \mid \mathcal{G}_{t-1} \right] \right) \\
&= \frac{1}{|A_{i,t}|} \left( \sum_{\ell \in A_{i,t-1}} M_{i,t}^{(\ell)} \cdot M_{i,\tau_{i,\ell}^{\mathrm{start}}-1} + \sum_{\ell \in A_{i,t} \setminus A_{i,t-1}} \mathbb{E}\left[ M_{i,t}^{(\ell)} \mid \mathcal{G}_{t-1} \right] \cdot M_{i,t-1} \right) \\
&\leq \frac{1}{|A_{i,t}|} \left( \sum_{\ell \in A_{i,t-1}} M_{i,t-1}^{(\ell)} \cdot M_{i,\tau_{i,\ell}^{\mathrm{start}}-1} + \sum_{\ell \in A_{i,t} \setminus A_{i,t-1}} M_{i,t-1} \right) \\
&= \frac{1}{|A_{i,t}|} \left( |A_{i,t-1}| \cdot M_{i,t-1} + |A_{i,t} \setminus A_{i,t-1}| \cdot M_{i,t-1} \right) \\
&= M_{i,t-1}.
\end{aligned}
$$

The sole inequality is because $(M_{i,t}^{(\ell)})$ is a supermartingale, and $M_{i,t}^{(\ell)} = 1$ when $t < \tau_{i,\ell}^{\text{start}}$. Thus, $\sup_{\tau \in \mathcal{T}} \mathbb{E}[E_{i,\tau}] \le \sup_{\tau \in \mathcal{T}} \mathbb{E}[M_{i,\tau}] \le 1$ where the final inequality is by optional stopping. Consequently, $(E_{i,t})$ are e-processes for $i \in [k]$ so $\sup_{\tau \in \mathcal{T}} \text{FDR}(\mathcal{S}_\tau) \le \delta$ by Fact 3, which achieves our desired result. $\qquad\square$

Nonnegative martingales play a central role in characterizing admissible e-processes — every e-process is upper bounded by a nonnegative martingale (Corollary 24; Ramdas et al. [31]). Thus, Proposition 13 proves that if $(E_{i,t}^{(\ell)})$ are all e-processes for $i \in [k], \ell \in \mathbb{N}$, then FDR control is maintained in the multi-agent for any stopping time.

### D.4 Hypotheses involving multiple arms

In the current setting, we have only considered hypotheses that are tied to a single arm i.e. hypothesis $i$ is concerned solely with $\nu_i$ for all $i \in [k]$. We also might be concerned with hypotheses that involve multiple arms. For example, we could be interested in the hypothesis that the reward distributions are exchangeable across arms [40, 42] i.e. any permutation of the arms is the same distribution, or the hypothesis that the means of two specific reward distributions are the same. Naturally, if each hypothesis is not restricted to being involved with only a single arm, we can consider more (or fewer) hypotheses than the number of arms.

Thus, we can denote $k$ to be the total number of hypotheses and $n$ to be the number of arms. Algorithm 6 specifies a meta-algorithm similar to Algorithm 1 that maintains FDR control in multi arm hypotheses. We simply maintain an e-process or p-process for each hypothesis. An important difference between hypotheses involving multiple arms setting and the standard setting is that the independence of $X_{1,t},...,X_{n,t}$ is no longer sufficient to ensure all the e-variables or p-variables are dependent only through the exploration policy and stopping time. The dependence structure within the e-variables or p-variables is based not only upon the dependence of $X_{1,t},...,X_{n,t}$, but also whether the hypothesis tests themselves have any dependence among each other e.g. two hypotheses might involve the same arm. Thus, for p-variables, we may require $\delta' = \delta/\log k$ even when the reward distributions are independent.

---

**Algorithm 6:** A meta-algorithm that ensures FDR control when hypotheses can involve multiple arms in the bandit setting.

---

**Input:** Exploration component $(\mathcal{A}_t)$, stopping rule $\tau^*$, desired level of FDR control $\delta \in (0,1)$. Set $D_0 = \emptyset$.

**for** $t$ *in* $1...$ **do**

    $\mathcal{I}_t := \mathcal{A}_t(D_{t-1}) \subseteq [n]$

    Obtain rewards for each $i \in \mathcal{I}_t$, and update data $D_t := D_{t-1} \cup \{(i,t,X_{i,t}) : i \in \mathcal{I}_t\}$.

    Update e-process or p-process that relate to any of the queried arms.

    $\mathcal{S}_t := \begin{cases} \text{BH}[\delta/\log k](p_{1,t},...,p_{k,t}) \text{ or arbitrary p-self-consistent set} & \text{if using p-variables} \\ \text{eBH}[\delta](e_{1,t},...,e_{k,t}) \text{ or arbitrary e-self-consistent set} & \text{if using e-variables} \end{cases}$

    **if** $\tau^* = t$ **then** stop and **return** $\mathcal{S}_t$;

**end**

---

**Proposition 14.** *Algorithm 6 outputs $\mathcal{S}_t$ for all $t \in \mathbb{N}$ such that $\sup_{\tau \in \mathcal{T}} \text{FDR}(\mathcal{S}_\tau) \le \delta$.*

E-variables in this setting have potentially larger power over p-variables than in the standard setting. This is because the number of hypotheses, $k$, is no longer tied to the number of arms, $n$. For example, $k \approx n^2/2$ if there was a hypothesis for each pair of arms in the bandit. Then, using p-variables in Algorithm 6 would require a correction of approximately $2 \log k$. In contrast, p-variables and BH require no more than a $\log k$ correction in the standard setting. Consequently, allowing for multiple arm hypotheses further highlights the benefit of e-variables over p-variables when dealing with arbitrarily dependent statistics.

## E  Additional simulations

In this section, we perform additional simulations to empirically verify our theoretical results. We test the performance of different choices of p-variables against e-variables in the standard bandit setting. We also provide simulations for the combinatorial bandit setting and compare p-variable methods with different assumptions against an e-variable method.

## E.1    Testing against different choices of p-variables

We consider two additional choices of p-variables to compare with our e-variable method and the p-variable from JJ discussed in Section 5. One is simply $\hat{P}_{i,t}^{\text{IPM-H}} := 1/E_{i,t}^{\text{PM-H}}$, which we will call Inverse PM-H (IPM-H). The other, which we call the IS p-variable, which is defined as follows by setting $\varphi = \varphi^{\text{IS}}$ in (5b).

$$P_{i,t}^{\text{IS}} := \inf\{\beta \in [0,1] : |\hat{\mu}_{i,t} - \mu_0| > \varphi^{\text{IS}}(t,\beta)\}. \tag{15}$$

We run these methods using the UCB arm selection algorithm described in (5a) inside of Algorithm 1.

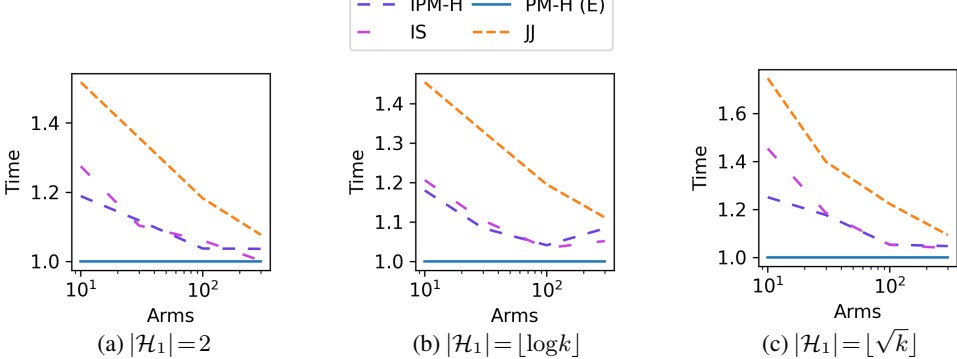

(a) $|\mathcal{H}_1| = 2$          (b) $|\mathcal{H}_1| = \lfloor \log k \rfloor$          (c) $|\mathcal{H}_1| = \lfloor \sqrt{k} \rfloor$

Figure 5: Relative comparison of time $t$ for each method to obtain a rejection set, $\mathcal{S}_t$, that has a $\text{TPR}(\mathcal{S}_t) \geq 1-\delta$ while maintaining $\text{FDR}(\mathcal{S}_t) \leq \delta$, where we choose $\delta = 0.05$. This plot compares different choices of p-variables against the PM-H e-variable over different numbers of arms (choices of $k$) and different densities of non-null hypotheses (sizes of $\mathcal{H}_1$). Time is reported as a ratio to the time taken by the algorithm that uses the PM-H e-variable. The JJ p-variable is the baseline p-variable specified in (5). We see that both the IPM-H and the IS p-variable have similar performance, and require fewer samples than the JJ p-variable. Overall, the PM-H e-variable performs better than any choice of p-variable.

The results shown in Figure 5 demonstrate that e-variables and e-BH still perform better than any p-variable and BH method. The two new p-variables, IS and IPM-H, have about similar sample efficiency, and both outperform the JJ p-variable, but both are still slightly worse than the PM-H e-variable. Thus, e-BH and e-variables have consistently better performance than BH and p-variables.

## E.2    Graph bandits with dependent $X_{1,t},...,X_{k,t}$

We consider a graph bandit setting where the algorithm makes no assumptions about the underlying dependence structure, and each arm consists of a node and its neighbors. We set the joint distribution over rewards at each step as the product of independent normal distributions for each arm. The marginal distribution of each arm $i \in [k]$ is a normal distribution with mean $\mu_i$, where $\mu_i = 1/2$ if $i \in \mathcal{H}_1$ and $\mu_i = \mu_0 = 0$ if $i \in \mathcal{H}_0$. Each graph we simulate is composed of 10 cliques of $k/10$ nodes. Thus, the set of superarms available for sampling is $\mathcal{K} = \{\{i, i+10, i+20, ..., i+k-10\} : \text{for } i \in [10]\}$. Finally, we let $\delta = 0.05$ be level of $\text{FDR}$ control for each algorithm.

We compare 3 different methods. For all 3 methods, the exploration strategy is to uniformly sample from the set of superarms $\mathcal{K}$. These methods differ solely in their choice of the evidence component. The first method is called the *single arm BH* method, as it only saves a single uniformly random sample from the set of samples it attains at each time step. Hence, it is equivalent to the uniformly randomly sampling BH method for the standard bandit setting. In this combinatorial bandit setting, it simply discards all but one sample at each step, and can consequently still enjoy the guarantees in Proposition 1. Our second method is to use the default BH and p-variables with no discarding of samples and the larger correction from Proposition 2. Lastly, we have the e-BH and e-variable method that also uses all samples from each pull of a superarm, since e-BH requires no correction for arbitrary dependence.

Figure 6 shows the results of using methods that guarantee $\text{FDR}$ control at level $\delta$ on graph bandits with arbitrary dependence between arms. Single arm BH pays a tremendous cost in time by throwing away many samples at each step, and the slightly smaller correction it needs to make does not make up for this deficit. Between the two methods that make full use of the samples obtained from superarm, we see that e-BH does better. Thus, e-variables and e-BH exhibit empirical performance on par or better than p-variables and BH in both the standard and combinatorial bandit settings.

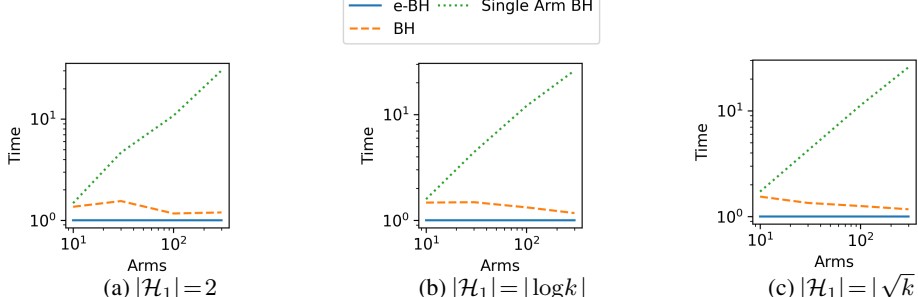

(a) $|\mathcal{H}_1| = 2$        (b) $|\mathcal{H}_1| = \lfloor \log k \rfloor$        (c) $|\mathcal{H}_1| = \lfloor \sqrt{k} \rfloor$

Figure 6: Relative comparison of time $t$ for each method to obtain a rejection set, $\mathcal{S}_t$, that has a $\mathrm{TPR}(\mathcal{S}_t) \geq 1 - \delta$ while maintaining $\mathrm{FDR}(\mathcal{S}_t) \leq \delta$, where we choose $\delta = 0.05$. This plot compares two different p-variable methods (BH and single arm BH) against an e-variable method (e-BH) over different numbers of arms (choices of $k$) and different densities of non-null hypotheses (sizes of $\mathcal{H}_1$). Time is reported as a ratio to the time taken by the algorithm that uses e-BH. We see that e-BH outperforms the two other BH algorithms in the graph bandit setting. Notably, single arm BH is linearly increasing in time relative to the other two methods that make full use of the samples obtained from a superarm. Single arm BH discards too many samples at each step, and the smaller correction it makes does not make up the deficit in number of samples.

## F   Betting interpretation of e-variables for bandits

We will describe our methodology for constructing e-variables using the perspective of betting in this section. Shafer [33] uses betting to formulate a paradigm for understanding the quantity represented by an e-value, and Shafer and Vovk [34] extend these ideas to form a mathematically rigorous foundation for probability based on game theory. Separately, betting ideas have also been used in parameter free techniques for online learning [28, 21, 29]. In this section, we will use a betting approach to produce a data adaptive e-process.

Recall that if $E$ is an e-variable, then $\mathbb{E}[E] \leq 1$ when the null hypothesis is true. On the other hand, if the null hypothesis is false, we would like $E$ to be large, since that increases the likelihood that the null hypothesis is rejected. Thus, constructing $E$ such that is satisfies the e-variable constraint under the null and is large under the alternative is the same as constructing a valid hypothesis test that has as much power as possible. Consequently, we can consider a betting game where we pay a dollar to play, and $E$ is the payout. If the null hypothesis is true, then we are unable to make any money in expectation, since the expectation is of $E$ is at most 1. However, if the null hypothesis is false, then we would expect to be able to make money on this game. If we did not make money under the alternative, then any test that used this e-variable would have no power, since the behavior of $E$ would not change between the null hypothesis being true and being false. In other words, this would be no better than picking $E = 1$ deterministically: a valid e-variable, but ineffectual for testing.

We define the **predictably-mixed Hoeffding (PM-H)** e-process [45], which we used in our simulations in Section 5, as follows:

$$E_{i,t}^{\mathrm{PM\text{-}H}}(\mu_0) := \prod_{j=1}^{T_i(t)} \exp(\lambda_{i,t_i(j)}(X_{i,t_i(j)} - \mu_0) - \lambda_{i,t_i(j)}^2/2).$$

$(\lambda_{i,t})$ is any sequence of nonnegative real numbers that is predictable w.r.t. $(\mathcal{F}_t)$. We will use an argument based on betting to derive a $(\lambda_{i,t})$ sequence, and show that this e-process can "make money" and hence provide TPR guarantees in the sub-Gaussian case. We first observe the following property of this process.

**Proposition 15.** $E_{i,t}^{\mathrm{PM\text{-}H}}(\mu_0)$ *is a nonnegative supermartingale, and thus an e-process, if $i \in \mathcal{H}_0$ and $\nu_i$ is 1-sub-Gaussian.*

*Proof.* We drop $\mu_0$ from $(E_{i,t}^{\mathrm{PM\text{-}H}}(\mu_0))$ and denote it as $(E_{i,t}^{\mathrm{PM\text{-}H}})$.

We proceed by showing $(E_{i,t}^{\mathrm{PM\text{-}H}})$ is a nonnegative supermartingale w.r.t. to the canonical filtration $(\mathcal{F}_t)$.

Consider $E_{i,t}^{\text{PM-H}}$ when $i \in \mathcal{H}_0$. If $I_t \neq i$ then $E_{i,t}^{\text{PM-H}} = E_{i,t-1}^{\text{PM-H}}$, which satisfies the supermartingale property in (6).

Otherwise,

$$
\begin{aligned}
\mathbb{E}[E_i^{\text{PM-H}}i,t \mid \mathcal{F}_{t-1}] &= \mathbb{E}\left[\exp\left(\lambda_t(X_{i,t} - \mu_0) - \frac{\lambda_t^2}{2}\right) E_i^{\text{PM-H}}i, t-1 \mid \mathcal{F}_{t-1}\right] \\
&= \mathbb{E}\left[\exp\left(\lambda_t(X_{i,t} - \mu_0) - \frac{\lambda_t^2}{2}\right) \mid \mathcal{F}_{t-1}\right] E_{i,t-1}^{\text{PM-H}} \\
&\leq E_{i,t-1}^{\text{PM-H}},
\end{aligned}
$$

where the final equality is because $X_{i,t}$ are independent across $t \in \mathbb{N}$ and 1-sub-Gaussian, and $\mu_i \leq \mu_0$.

Since $(E_{i,t}^{\text{PM-H}})$ is a nonnegative supermartingale, it is an e-process by optional stopping. Thus, we have achieved our desired result. $\qquad\square$

Note that Proposition 15 justifies that our choice of e-process for the simulations in Section 5 was indeed a valid e-process. Now that we have shown $(E_{i,t}^{\text{PM-H}})$ is an e-process, we will consider how to choose a powerful $(\lambda_{i,t})$. Consider a model where we view the e-value, $e_{i,t}$, for each arm $i \in [k]$ as the money made by each arm, or a "**betting score**". For each arm $i \in [k]$, imagine we are allocated initial wealth equal to 1. At each time step, the algorithm chooses an arm $i \in [k]$, and a "bet", $\lambda_{i,t}$. The wealth of the arm at the next round changes by a factor based on the reward $X_{i,t}$ (assuming $i$ is the arm chosen at round $t+1$):

$$
e_{i,t+1} = e_{i,t} \cdot \underbrace{\exp(\lambda_{i,t}(X_{i,t} - \mu_0) - \lambda_{i,t}^2/2)}_{\text{change in wealth}}.
$$

Note that this a "fair game" or the reward multiplier is less than 1 in expectation if $\mathbb{E}[X_{i,t}] \leq \mu_0$.

The betting score, $E_{i,t}^{\text{PM-H}}$, may be interpreted as the money earned by arm $i$ at time $t$. When the null hypothesis is true, i.e. $\mu_i \leq \mu_0$, we know that $\sup_{\tau \in \mathcal{T}} \mathbb{E}[E_{i,\tau}^{\text{PM-H}}] \leq 1$ by Proposition 15. Thus, regardless of our stopping strategy, we make no money in expectation. However, if we knew that $E_{i,t}^{\text{PM-H}}$ was actually a favorable bet, and $\mathbb{E}[X_{i,t}] = \mu_i > \mu_0$, we would want to come up with a sequence $(\lambda_{i,t})$ for each arm $i \in [k]$ that maximizes our wealth at each arm. Consequently, we can reframe our goal for choosing $(\lambda_{i,t})$ as maximizing capital in a betting game. In the next section, we will discuss some strategies for accomplishing such an objective.

### F.1 Optimal betting strategies

One way of maximizing capital is to optimize for the Kelly criterion [24], which aims to maximize the logarithm of the capital on each step and is equivalent to maximizing rate of growth of capital. In our scenario, the Kelly criterion manifests in the following form:

$$
\mathbb{E}\left[\log E_{i,t}^{\text{PM-H}}(\mu_0)\right] = \sum_{j=1}^{T_i(t)} \mathbb{E}\left[\lambda_{i,t_i(j)}(X_{i,t_i(j)} - \mu_0) - \lambda_{i,t_i(j)}^2/2\right].
$$

**Optimal choice of $(\lambda_t)$ for log wealth.** To maximize the above sum, we can simply decompose it with respect to each $j$, and since the $\lambda_{i,t_i(j)}$ are decoupled, we can identify an optimal $\lambda_{i,t_i(j)}^*$ for each $j$:

$$
\lambda_{i,t_i(j)}^* := \operatorname{argmax}_{\lambda \in \mathbb{R}^+} \lambda \mathbb{E}[X_{i,t_i(j)} - \mu_0] - \lambda^2/2 = \mu_i,
$$

$$
\lambda_{i,t_i(j)}^* \mathbb{E}[X_{i,t_i(j)} - \mu_0] - {\lambda_{i,t_i(j)}^*}^2/2 = \max_{\lambda \in \mathbb{R}^+} \lambda \mathbb{E}[X_{i,t_i(j)} - \mu_0] - \lambda^2/2 = \Delta_i^2/2.
$$

We can see that if the $\mu_i$ is known, the above quantity is maximized by setting $\lambda_{i,t_i(j)} = \mu_i$ for all $j \in [T_i(t)]$. This observation confirms our intuition that the Kelly criterion is a sensible quantity to optimize for when trying to maximize the e-values of hypothesis in $\mathcal{H}_1$. On the other hand, if $i \in \mathcal{H}_0$, $\mu_i \leq \mu_0 = 0$, the log wealth incurred at each time step is nonpositive. Thus, in expectation, the log wealth process $\log E_{i,t}^{\text{PM-H}}(\mu_0)$ will only increase in capital when the hypothesis associated with the

arm is truly non-null. In betting language, we are presenting a one-sided bet that allows for our bets $(\lambda_{i,t})$ to make money in expectation iff the null hypothesis is false. Hence, our strategy is profitable only when the true mean of the arm is greater than $\mu_0$.

In practice, we do not know $\mu_i$, since testing $\mu_i$ is the entire premise of the problem. Instead, we can use the sample mean, $\widehat{\mu}_{i,t}$, in place of $\mu_i$ and show that it gives us convergence at a rate of approximately $1/T_i(t)$ to the optimal capital gain rate.

**Proposition 16.** *Let $\nu_i$ be $1$-sub-Gaussian for $i \in [k]$. If $\lambda_{i,t} = \hat{\mu}_{i,t-1}$, then*

$$\mathbb{E}[\widehat{\mu}_{i,t_i(j)-1}(X_{i,t_i(j)} - \mu_0) - \widehat{\mu}_{i,t_i(j)-1}^2/2] = \Delta_i^2/2 - 1/(T_i(t)-1).$$

Proposition 16 follows from the variance of $\widehat{\mu}_{i,t}$ being $1/T_i(t)$. Now, we can derive the following corollary.

**Corollary 1.** *The total log wealth at time $t$, $\log E_{i,t}^{\mathrm{PM\text{-}H}}$, has an expectation satisfying the following property, where $\lambda_{i,t} = \widehat{\mu}_{i,t-1}$:*

$$\mathbb{E}[\log E_{i,t}^{\mathrm{PM\text{-}H}}] = T_i(t)\Delta_i^2/2 - \sum_{j=1}^{T_i(t)} 1/j \approx T_i(t)\Delta_i^2/2 - \log(T_i(t)),$$

*where $\sum_{j=1}^{T_i(t)} 1/j$ is approximately $\log(T_i(t))$.*

Thus, in log wealth, using $\widehat{\mu}_{i,t}$ incurs a penalty of $\log T_i(t)$, which is relatively small compared to the positive term — especially when $t$ is large.

## F.2 Sample complexity for standard sub-Gaussian bandits

We prove a sample complexity result for the $E_{i,t}^{\mathrm{PM\text{-}H}}$ as well.

**Theorem 2.** *Let $(\mathcal{A}_t)$ be such that $\mathcal{A}_t$ outputs $\mathcal{I}_t = \{I_t\}$ for all $t \in \mathbb{N}$, where $I_t$ is defined in (5a), $\lambda_{i,t} = (\widehat{\mu}_{i,t-1}/2)_+$, and $E_{i,t} = E_{i,t}^{\mathrm{PM\text{-}H}}$. Then, Algorithm 1 will always guarantee $\sup_{\tau \in \mathcal{T}} \mathrm{FDR}(\mathcal{S}_\tau) \leq \delta$. With at least $1-\delta$ probability, there will exist*

$$T \lesssim \sum_{i \in \mathcal{H}_0} \Delta_i^{-2} \log(\log(\Delta_i^{-2})/\delta) + \sum_{i \in \mathcal{H}_1} \Delta_i^{-2}(\log(\Delta_i^{-2})\log(1/\delta) + \log k)$$

$$\wedge |\mathcal{H}_0|\Delta^{-2}\log(\log(\Delta^{-2})/\delta) + |\mathcal{H}_1|\Delta^{-2}\log(\Delta^{-2})\log(1/\delta)$$

*such that $\mathrm{TPR}(\mathcal{S}_t) \geq 1-\delta$ for all $t \geq T$.*

Theorem 2 shows the limitation of using an estimate of the mean, $\widehat{\mu}_{i,t}$, in place of the true mean. The intuition of the proof of Theorem 2 is that at each step, $E_{i,t}^{\mathrm{PM\text{-}H}}$ must account for an $1/t$ deviation, since the variance of $\widehat{\mu}_{i,t}$ is $1/t$. The sum of these deviations is approximately $\log t$. Thus, the sample complexity bound has a $\log \Delta^{-2}$ instead of only a $\log\log \Delta^{-2}$ term. This limitation seems to be an inherent flaw in choice of $(\lambda_{i,t})$ based on estimation, since the estimation error must be accounted for along with the typical deviation from providing a concentration inequality that is uniform over time steps $t$.

To prepare for our proof of Theorem 2, we require some self-contained lemmata. Define the following auxiliary random variables for all $i \in \mathcal{H}_1$:

$$\rho_i' := \min_{t \in \mathbb{N}} \frac{E_{i,t}}{\exp\left(\sum_{j=1}^{T_i(t)} \lambda_{i,t_i(j)}\Delta_i - \lambda_{i,t_i(j)}^2\right)}. \tag{16}$$

**Lemma 7.** *For all $i \in \mathcal{H}_1$, $\mathbb{P}(\rho_i' \leq s) \leq s$ for $s \in (0,1)$ i.e. $\rho_i'$ is superuniformly distributed.*

*Proof.* We observe that the reciprocal of $\rho_i'$ is the following:

$$1/\rho_i' = \max_{t \in \mathbb{N}} \exp\left(\sum_{j=1}^{T_i(t)} \lambda_{i,t_i(j)}(\mu_i - X_{i,t_i(j)}) - \lambda_{i,t_i(j)}^2/2\right).$$

Let

$$M_t = \exp\left(\sum_{j=1}^{T_i(t)} \lambda_{i,t_i(j)}(\mu_i - X_{i,t_i(j)}) - \lambda_{i,t_i(j)}^2/2\right).$$

We will show $(M_t)$ is a nonnegative supermartingale w.r.t. $(\mathcal{F}_t)$. Assume arm $i$ is sampled at time $t$ — otherwise the supermartingale property is trivially satisfied.

$$\mathbb{E}[M_t \,|\, \mathcal{F}_{t-1}] = \mathbb{E}\left[\exp\left(\sum_{j=1}^{T_i(t)} \lambda_{i,t_i(j)}(\mu_i - X_{i,t_i(j)}) - \lambda_{i,t_i(j)}^2/2\right) \Big| \mathcal{F}_{t-1}\right]$$

$$= \mathbb{E}\left[\exp\left(\lambda_{i,t}(\mu_i - X_{i,t}) - \lambda_{i,t}^2/2\right) | \mathcal{F}_{t-1}\right] \exp\left(\sum_{j=1}^{T_i(t-1)} \lambda_{i,t_i(j)}(\mu_i - X_{i,t_i(j)}) - \lambda_{i,t_i(j)}^2/2\right)$$

$$\leq \exp\left(\sum_{j=1}^{T_i(t-1)} \lambda_{i,t_i(j)}(\mu_i - X_{i,t_i(j)}) - \lambda_{i,t_i(j)}^2/2\right)$$

$$= M_{t-1}.$$

The sole inequality arises from $X_{i,t}$ being independent across $t \in \mathbb{N}$ and 1-sub-Gaussian, and having mean $\mu_i$.

Thus, $\rho_i'$ is superuniformly distributed by Ville's inequality. $\qquad\square$

Rewriting the definition of $\rho_i'$, we get:

$$E_{i,t} \geq \exp\left(\sum_{j=1}^{T_i(t)} \lambda_{i,t_i(j)}\Delta_i - \lambda_{i,t_i(j)}^2\right)\rho_i' \tag{17}$$

for all $t \in \mathbb{N}$. Now show a result for the rate of growth of $E_{i,t}$ by showing a result concerning the lower bound in (17).

**Lemma 8.** *For all $t \in \mathbb{N}$,*

$$\exp\left(\sum_{j=1}^{T_i(t)} \lambda_{i,t_i(j)}\Delta_i - \lambda_{i,t_i(j)}^2\right) \gtrsim T_i(t)\Delta_i^2 - \log(1/\rho_i)\log(T_i(t)).$$

*Proof.* Recall that $\lambda_t = (\widehat{\mu}_{i,t-1}/2)_+$. Then, we derive the following asymptotic lower bound:

$$\sum_{j=1}^{T_i(t)} \lambda_{i,t_i(j)}\Delta_i - \lambda_{i,t_i(j)}^2 = \frac{1}{4}\sum_{j=1}^{T_i(t)} 2\widehat{\mu}_{i,t-1}\Delta_i - \widehat{\mu}_{i,j-1}^2$$

$$\geq \frac{1}{4}\sum_{j=1}^{T_i(t)} \Delta_i^2 - (\Delta_i - \widehat{\mu}_{i,j-1})^2$$

$$\geq \frac{1}{4}\sum_{j=1}^{T_i(t)} \Delta_i^2 - \varphi(T_i(j-1),\rho_i)^2 \qquad\qquad \text{def. of } \rho_i$$

$$\geq \frac{1}{4}\sum_{j=1}^{T_i(t)} \Delta_i^2 - \frac{4\log(\log_2(2j)/\rho_i)}{j} \qquad\qquad \text{upper bound from Fact 4}$$

$$\gtrsim \Delta_i^2 T_i(t) - \log\left(\frac{1}{\rho_i}\right)\log(T_i(t)),$$

where the last line is because $\sum_{j=1}^{T_i(T)} 1/j \approx \log T_i(t)$. Thus, we have arrived our desired result. $\qquad\square$

We now have the ingredients to present a proof of Theorem 2.

*Proof of Theorem 2.* Combining the lower bound in (17) with Lemma 8, we get the following asymptotic lower bound:

$$E_{i,t} \gtrsim \exp(T_i(t)\Delta_i^2 - \log(1/\rho_i') - \log(1/\rho_i)\log(T_i(t))).$$

Inverting the expression above, we get that the following lower bound sample complexity of a single arm,

$$T_i(t) \gtrsim \Delta_i^{-2}\log(\Delta_i^{-2})\log(1/\rho_i) + \Delta_i^{-2}\log(1/\rho_i') + \Delta_i^{-2}\log(\varepsilon),$$

implies $E_{i,t} \geq \varepsilon$ for $\varepsilon > 0$.

We can now derive a bound for $\sum_{t=1}^{\infty}\mathbf{I}\{I_t \in \mathcal{H}_1, \mathcal{R} \not\subseteq \mathcal{S}_t\}$.

$$\sum_{t=1}^{\infty}\mathbf{I}\{I_t \in \mathcal{H}_1, \mathcal{R} \not\subseteq \mathcal{S}_t\} \leq \max_{\pi}\sum_{i \in \mathcal{H}_1}\Delta_i^{-2}\log(\Delta_i^{-2})\log(1/\rho_i) + \Delta_i^{-2}\log(1/\rho_i') + \Delta_i^{-2}\log(k/\pi(i))$$

where $\pi$ is a mapping from $[|\mathcal{H}_1|]$ to $\mathcal{H}_1$.

We get the following total bound:

$$\begin{aligned}
\sum_{t=1}^{\infty}\mathbf{I}\{\mathcal{R} \not\subseteq \mathcal{S}_t\} &= \sum_{t=1}^{\infty}\mathbf{I}\{I_t \in \mathcal{H}_0, \mathcal{R} \not\subseteq \mathcal{S}_t\} + \sum_{t=1}^{\infty}\mathbf{I}\{I_t \in \mathcal{H}_1, \mathcal{R} \not\subseteq \mathcal{S}_t\} \\
&= \sum_{t=1}^{\infty}\mathbf{I}\{I_t \in \mathcal{H}_0, \mathcal{R} \not\subseteq \mathcal{S}_t\} + \sum_{t=1}^{\infty}\mathbf{I}\{I_t \in \mathcal{H}_1, \mathcal{R} \not\subseteq \mathcal{S}_t\} \\
&\lesssim \sum_{i \in \mathcal{H}_0}\Delta_i^{-2}\log(\log(\Delta_i^{-2})/\delta\rho_i) \\
&\quad + \max_{\pi}\sum_{i \in \mathcal{H}_1}\Delta_i^{-2}\log(\Delta_i^{-2})\log(1/\rho_i) + \Delta_i^{-2}\log(1/\rho_i') + \Delta_i^{-2}\log(k/\pi(i)),
\end{aligned}$$

where the asymptotic inequality is by Lemma 5.

We know that we can apply Lemma 4 at level $\beta = \delta/2$ to $\rho_i$ for $i \in [k]$ and $\rho_i'$ for $i \in \mathcal{H}_1$. Thus, the following happens with at least $1 - \delta$ probability:

$$\begin{aligned}
\sum_{t=1}^{\infty}\mathbf{I}\{\mathcal{R} \not\subseteq \mathcal{S}_t\} &\lesssim \sum_{i \in \mathcal{H}_0}\Delta_i^{-2}\log(\log(\Delta_i^{-2})/\delta) \\
&\quad + \max_{\pi}\sum_{i \in \mathcal{H}_1}\Delta_i^{-2}\log(\Delta_i^{-2})\log(1/\delta) + \Delta_i^{-2}\log(k/\pi(i)).
\end{aligned}$$

Similar to the Theorem 1, we can show two different bounds. The first is the following:

$$\sum_{t=1}^{\infty}\mathbf{I}\{\mathcal{R} \not\subseteq \mathcal{S}_t\} \lesssim |\mathcal{H}_0|\Delta^{-2}\log(\log(\Delta^{-2})/\delta) + |\mathcal{H}_1|\Delta^{-2}\log(\Delta^{-2})\log(1/\delta),$$

because $\sum_{i=1}^{|\mathcal{H}_1|}\log(k/i) \leq k$. The second follows from dropping $\pi(i)$:

$$\begin{aligned}
\sum_{t=1}^{\infty}\mathbf{I}\{\mathcal{R} \not\subseteq \mathcal{S}_t\} &\lesssim \sum_{i \in \mathcal{H}_0}\Delta_i^{-2}\log(\log(\Delta_i^{-2})/\delta) \\
&\quad + \sum_{i \in \mathcal{H}_1}\Delta_i^{-2}\log(\Delta_i^{-2})\log(1/\delta) + \Delta_i^{-2}\log k.
\end{aligned}$$

Thus, we have shown both of our desired bounds. □

# G Testing the average conditional mean

For simplicity, we will discuss results and proofs in this section under the single arm bandit case, so we will drop the arm index $i$ when labeling terms. Our conclusions, however, do generalize to the general multi-arm bandit case.

In Section 4 and JJ, the null hypothesis for each arm we are concerned with is

$$\text{``}\mathbb{E}[X_t \,|\, \mathcal{F}_{t-1}] \leq \mu_0 \text{ for all } t \in \mathbb{N} \text{ almost surely.''} \tag{H1}$$

In the aforementioned settings, there is an additional assumption that $X_{i,t}$ are i.i.d. across $t \in \mathbb{N}$. Thus, $\mathbb{E}[X_t \,|\, \mathcal{F}_{t-1}]$ simply becomes $\mathbb{E}[X_t]$. We can also test a more general hypothesis of whether the means of $X_t$ are less than or equal to $\mu_0$ on average.

To formally define a notion of "average mean", let us consider the case where there is a single arm i.e. we have a sequence of rewards $X_1, X_2, ...$, where the *average conditional mean* is defined as

$$\overline{\mu}_t \equiv \frac{1}{t} \sum_{j=1}^{t} \mathbb{E}[X_j \,|\, \mathcal{F}_{j-1}].$$

Consequently, we can define a null hypothesis w.r.t. $\overline{\mu}_t$:

$$\text{``}\overline{\mu}_t \leq \mu_0 \text{ for all } t \in \mathbb{N} \text{ almost surely.''} \tag{H2}$$

In the specific case where $X_t$ are i.i.d. across $t \in \mathbb{N}$, each with mean $\mu$, then $\mathbb{E}[X_t \,|\, \mathcal{F}_{j-1}] = \mathbb{E}[X_t] = \mu$ for all $t \in \mathbb{N}$, and $\overline{\mu}_t = \mu$. Consequently, there would be no difference between testing the average conditional mean and testing the marginal mean, $\mu$, because they are the same value. However, when the distribution of $X_t$ are not necessarily i.i.d. across $t \in \mathbb{N}$, we will emphasize that not all valid tests for (H1) are also valid for (H2). Generally, (H1) is a "stronger" hypothesis than (H2) in the sense that any distribution over $X_t$ for $t \in \mathbb{N}$ that satisfies (H1) also satisfies (H2).

The difference between (H1) and (H2) is reflected in the fact that e-processes are supermartingales in (H1), but only upper bounded by a martingale in (H2).

**Proposition 17.** *Assume that conditional distribution of $X_t \,|\, \mathcal{F}_{t-1}$ is always $1$-sub-Gaussian for all $t \in \mathbb{N}$. Consider a process of the form,*

$$E_t := \sum_{\ell=1}^{m} w_\ell \exp\left( \sum_{j=1}^{t} \lambda_\ell (X_j - \mu_0) - \frac{\lambda_\ell^2}{2} \right)$$

*where $m \in \mathbb{N} \cup \{\infty\}$, $\sum_{\ell=1}^{m} w_\ell \leq 1$. Under both (H1) and (H2), $(M_t)$ is a e-process. Specifically, $(M_t)$ is*

 *(i) a nonnegative supermartingale under (H1).*

 *(ii) upper bounded by a nonnegative supermartingale under (H2).*

*Proof.* (i) follows from Proposition 7.

Without loss of generality, we will consider the case where $m = 1$ and $w_1 = 1$, since a convex combination of supermartingales is a supermartingale. Thus,

$$E_t = \exp\left( \sum_{j=1}^{t} \lambda(X_j - \mu_0) - \frac{\lambda^2}{2} \right).$$

To prove (ii), we first notice we can define a process $M_t'$ that upper bounds $E_t$:

$$M_t' = \exp\left(\sum_{j=1}^{t}\lambda(X_j - \mathbb{E}[X_j \mid \mathcal{F}_{j-1}]) - \frac{\lambda^2}{2}\right)$$

$$= \exp\left(\lambda\left(\sum_{j=1}^{t}X_j - \sum_{j=1}^{t}\mathbb{E}[X_j \mid \mathcal{F}_{j-1}]\right) - \frac{t\lambda^2}{2}\right)$$

$$= \exp\left(\lambda\left(\sum_{j=1}^{t}X_j - t\overline{\mu}_t\right) - \frac{t\lambda^2}{2}\right)$$

$$\geq \exp\left(\lambda\left(\sum_{j=1}^{t}X_j - t\mu_0\right) - \frac{t\lambda^2}{2}\right)$$

$$= E_t.$$

Now, we will show that $(M_t')$ is a supermartingale.

$$\mathbb{E}\left[\exp\left(\sum_{j=1}^{t}\lambda(X_j - \mathbb{E}[X_j \mid \mathcal{F}_{j-1}]) - \frac{\lambda^2}{2}\right) \mid \mathcal{F}_{t-1}\right] = \mathbb{E}\left[\exp\left(\lambda(X_t - \mathbb{E}[X_t \mid \mathcal{F}_{t-1}]) - \frac{\lambda^2}{2}\right) \mid \mathcal{F}_{t-1}\right]M_{t-1}'$$

$$\leq M_{t-1}',$$

where the last inequality is because the conditional distribution of $X_t \mid \mathcal{F}_{t-1}$ is 1-sub-Gaussian. Thus, we have shown both parts of our desired result. $\square$

The $E_t$ specified in Proposition 17 is an e-process, but not necessarily a nonnegative supermartingale, for any distribution under (H2) where there exists a $t \in \mathbb{N}$ such that $\mathbb{E}[X_t \mid \mathcal{F}_{t-1}] > \mu_0$. Thus, the distinction highlighted in Proposition 17 is not vacuous.

Further, we will also note the following negative result that there exists processes that are e-processes under (H1) but are not under (H2).

**Proposition 18.** *Assume that conditional distribution of $X_t \mid \mathcal{F}_{t-1}$ is always 1-sub-Gaussian for all $t \in \mathbb{N}$. $(E_t^{\mathrm{PM\text{-}H}})$ is an e-process under all distributions satisfying (H1), but there exist $(\lambda_t)$ such that $(E_t^{\mathrm{PM\text{-}H}})$ is not an e-process under all distributions that satisfy (H2).*

*Proof.* $(E_t^{\mathrm{PM\text{-}H}})$ is an e-process under (H1) by a similar argument to the proof of Proposition 15, since the conditional distribution of $X_t \mid \mathcal{F}_{t-1}$ is 1-sub-Gaussian. However, we can provide a simple counterexample choice of $(\lambda_t)$ and distribution that satisfies (H2) which cannot have expectation greater than 1 at a time $t \in \mathbb{N}$. Let $\mu_0 = 0$, $X_t = -1$ if $t$ is odd, and $X_t = 1$ if $t$ is even. Consider a $(\lambda_t)$ where $\lambda_t = 0$ when $t$ is odd and $\lambda_t = 1$ when $t$ is even. Then,

$$\mathbb{E}[E_t^{\mathrm{PM\text{-}H}}] = \exp(\lceil t/2\rceil).$$

Consequently, $(E_t^{\mathrm{PM\text{-}H}})$ with this choice of $(\lambda_t)$ is not an e-process under (H2), and we have proved our desired result. $\square$

Thus, using adaptive strategies for selecting $(\lambda_t)$ like in Waudby-Smith and Ramdas [45] for testing (H2) is not necessarily straightforward, while mixture strategies in the form specified in Proposition 17 are valid e-processes for testing both (H1) and (H2).