# OpenReview forum: "A unified framework for bandit multiple testing"
_NeurIPS.cc/2021/Conference — NeurIPS 2021 Poster_

### Official Review · Reviewer_MD8E · 2021-07-01

**Rating:** 5
**Confidence:** 3

**Summary:**

This paper studies the problem of Bandit Approach to Multiple Testing.

It shows the advantage of e-variables against p-variables in the use of multiple testing by proposing the following:
- extends the original problem to a combinatorial problem where a player can pull multiple arms
- a meta algorithm, adapted for both p-variables and e-variables
- comparing the confidence levels to use for both methods to reach an FDR < alpha
- comparing more specifically the performances of the original instance of multi-armed bandits (non-combinatorial)

**Limitations And Societal Impact:**

See above for the limitations

Societal impact: not concerned

**Main Review:**

I have read the other reviews and authors' answers. The authors did not totally dismiss my concerns. In particular, it seems that the use of e-processes is mostly beneficial in the specific case of combinatorial sampling (with dependent arms), which is too quickly mentioned in the paper (not counting the appendix). I think this aspect should be more showcased in the general structure of the paper to nicely illustrate the advantage of e-processes over p-processes with bandits.

--------------------------------------------------

This paper shows the interest of using e-variables against p-variables in multiple testing (with a bandits approach). I think this is a nice problematic to consider and that it indeed brings improvements to the current used schemes (in a black box manner).
Although, the paper seems poorly written overall and I think it requires to be rewritten in a more comprehensive/concise/accurate way.

The paper introduces a combinatorial version of the bandits hypothesis testing problem and the presented results (see eg Fact 2) seem to insist on the benefits of e-variables for this specific case. Yet the theoretical results of Section 6 and the experiments of Section 7 only consider the original single arm bandit (although there are expe for combinatorial in appendix), which limits the improvements brought by the use of e-variables. I think the paper should either focus on the sole problem of a single pull per round or the combinatorial case for clarity and simplicity. This will also lighten the writing, since it is overall very dense: we feel that the authors struggled to fit the 9 pages limit (see for example the page 8 with equations in the middle of the text).
Moreover, there is no mention of the literature on combinatorial bandits, which would provide first insights on the exploration rules to use in these cases.

When claiming the Facts 1,2,3: we do not know if the difference in the used levels of confidence come from the analysis (that might not be tight), or if we indeed have examples where you cannot use smaller levels. In particular for the Fact 2, do we have an example of dependencies where we indeed have FDR(S) of order alpha(1+log(1/alpha))log(k) ?

Although the improvement of the e-variables is well illustrated by the different levels of confidence used, it seems that it will only imply a reduction by a constant factor in the sample complexity of the used processes. Indeed, note that the dependence in delta is often ln(1/delta) and changing delta by delta/ln(1/delta) will not considerably change this value in the end. This largely limits the interest of e-variables as it will only reduce the sample complexity by a constant factor. Moreover, this factor seems to reduce to 1 when the number of arms grows as can be shown by the experiments.
Because of this, removing the constant terms in the Theorems (eg Thm 1) hides the real improvement of e-variable: it actually seems that there is no theoretical improvement at all.

In section 6, the UCB exploration in multiple testing seems incorrect: JJ does not take the argmax over [K] but over [K]\S_t
Moreover, the use of the parameter R does not seem really fair when comparing with the algo of JJ that does not use it. Especially, the authors need to know Delta to correctly tune R to then get their bound. Moreover, R leads to a uniform exploration instead of an adaptive one (which is why we have k /Delta^2 instead of sum 1/Delta_i^2 in the sample complexity given by Theorem 1.
Thus, I do not understand the interest of R at all, as I would have expected the algorithm to work even with R=0.
Because of this additional R, it even becomes unclear what algorithms are exactly compared in Figure 2. Does UCB BH corresponds to equation (3) and UCB e-BH to equation (4)? Then these are not the same exploration rules, so this is not a fair comparison as explained above.

------ minor comments ------

There are also many typos.

The notation k for the number of arms is not really the best (for example you have a contradicting notation in Appendix E because of this).

**Time Spent Reviewing:**

3 hours

---

> ### Author Response · Authors · 2021-08-10
> **Response to Reviewer MD8E**
>
> **Combinatorial and single arm settings**: Our inclusion of the combinatorial and single arm settings is to highlight that our framework matches the performance of existing bandit multiple testing methods which have only considered the independent, single arm case, and also generalizes to multiple testing in the combinatorial, dependent case. The power of our framework comes from its applicability to any bandit setting (and some more general settings discussed in Appendix B with multiple agents, multiple arms per hypothesis, etc.). We have updated the draft to include references to existing exploration strategies in the combinatorial bandits (which we do not explicitly rely on since we are agnostic to the exploration strategy, but are useful references anyway for good exploration strategies in non-multiple-testing contexts), and cleaned up the presentation of our techniques.
>
> **Tightness of FDR bounds**: The bound in Fact 2 for p-compliance on dependent p-variables is the best known upper bound from Su (2018), and is the only potentially non-tight bound (i.e, it might be tight or loose, but it is not currently known which). All other bounds (including p-compliance on independent p-variables) are tight i.e. there exist p-variable or e-variable distributions that have an FDR which is arbitrarily close or exactly equal to the FDR dictated by the upper bound. We have made these points clear in the final draft.
>
> **Sample complexity of e-variables**: We agree that, in the case where rewards of each arm are *independent*, e-variables do not improve sample complexity by more than a constant factor over p-variables. However, under arbitrary dependence, e-variables improve FDR by a factor of $\log k$ over p-variables, and consequently do not have FDR affected by the number of arms. This makes the improvement of e-variables over p-variables quite significant, since the FDR for e-variables no longer has any reliance on $k$. Thus, our goal with Theorem 1 is not to demonstrate that e-variables outperform p-variables in the independent, single-arm bandit setting. Instead, we show that it can achieve the same sample complexity guarantees in settings considered in prior work, and is more powerful in arbitrary dependence settings (while being valid without any change to the method, regardless of problem setup).
>
> **UCB algorithm with e-variables**: We show in the appendix that e-variables can achieve matching sample complexity with p-variables under the UCB algorithm (see root level comment). We have fixed the typo for our formulation of UCB in the final draft. Since we can directly use UCB, we no longer require knowledge of $\Delta$ or have any parameters like $R$ to tune. For the numerical simulations, we state in Section 7 under "Simulation setup" that we use the same exploration strategies for both p-variables and e-variables - we test uniform sampling and the UCB sampling method specified in (3a). Thus, we do compare p-variables and e-variables using the same two exploration rules in our simulations. We have clarified our setup description to indicate this more explicitly in our final draft.
>
> **Minor comments**: Thank you for highlighting this inconsistency - we have corrected the notation.

---

### Official Review · Reviewer_7wSu · 2021-07-13

**Rating:** 5
**Confidence:** 4

**Summary:**

The authors give a unified framework for bandit multiple testing in terms of e-values. This framework is more flexible than prior work, allowing for dependencies between random variables and adaptive algorithm without corrections required in prior work, most notably JJ. The authors apply this result in a setting studied previouly where each arm is sub-gaussian and independent, and a single arm is queried at each step. They give sample complexity bounds matching prior state-of-the-art and show comparable performance in experiments.

**Limitations And Societal Impact:**

I do not see any potential negative societal impact.

**Main Review:**

The framework of e-processes seems quite useful for bandit multiple testing and this observation seems very important.

It is not clear to me what the technical contributions of the current paper are over prior work on e-processes (e.g., Wang and Rambdas, 2020, and Waudby-Smith and Ramdas, 2021). Is the main contribution the usefulness of this framework to bandit multiple testing? Or, are there other technical challenges that had to be overcome to apply this framework to bandit multiple testing?

It seems that the algorithm in Section 6 using e-processes requires knowledge of Delta through the parameter R, which is often not known in practice. Is knowledge of Delta required to use this algorithm? If it is, then this would make the approach less practical. It may be that some sort of doubling method can be applied, but in that case would the guarantee still match the prior-state-of-the-art and would the algorithm perform as well in practice?

The authors write that the Theorem 1 can likely be generalized to the setting where the Delta_i have very different values. Given the use of the warm-start to make \hat{\mu}_{i,t} a good estimate of \mu_i, I don't see what this algorithm would look like. It would be helpful to elaborate on this point since if it is difficult to prove bounds depending on Delta_i in this framework, that would make it less theoretically appealing.

The authors compare against BH in the setting where Delta_i = Delta for all i. It would be useful to see a comparison for the setting where the Delta_i's do not vary. Thus, we could at least get a sense of the performance of e-BH in this setting. Since this setting is common, it would be relevant to practice.

I think that it would be useful to move some of the other problem settings simulations from the appendix to the main paper. This would help clarify the advantage of flexibility of UCB e-BH over UCB BH and make the algorithms more concrete.

After discussion: after reading the authors' responses and the other reviews, I maintain my score.


**Time Spent Reviewing:**

4

---

> ### Author Response · Authors · 2021-08-10
> **Response to Reviewer 7wSu**
>
> **Technical contributions of this work**: We agree that one main contribution is to show the usefulness of e-processes and e-BH (both very new concepts) in bandit multiple testing. Another contribution of this paper is the recognition that the exploration and evidence components can be decoupled. This decoupling, although possible for p-variables at a price, is more applicable to e-variable due to their robustness to dependency. The decoupling is what allows us to apply the framework to a wide variety of settings, since we only need to show the evidence component always maintains valid e-processes.
> Last, since e-processes are e-variables at arbitrary stopping times, the rejection set output by e-BH when the bandit algorithm terminates has valid FDR control. JJ proved the validity of FDR guarantees for p-variables by using the fact that the limiting infimum of a p-process is a p-variable. However, this fact is not true for e-variables i.e. the supremum of an e-process is not an e-variable. Hence, validity of FDR control for e-BH and e-variables arises from a different proof. On the relationship to other mentioned past work, we also note that prior work did not consider multiple testing and adaptive exploration strategies together for e-variables. Wang and Ramdas (2020) did not consider exploration over multiple arms at all --- they only proved properties of e-BH when provided with an e-variable for each hypothesis.
> In contrast, Waudby-Smith and Ramdas (2021) only provide e-processes for testing a hypothesis on bounded distributions and are not concerned with multiple testing, or multiple arms. Thus, our contribution is to recognize that is e-BH is valid in the multi-arm bandit setting and provides robustness to different types of dependence --- we have delineated these points more in the final paper.
>
> **Usability of Theorem 1**: We have formulated an improved algorithm that matches sample complexity bounds with the p-variable method in JJ (see root level comment). This algorithm does not require any prior knowledge or $\Delta_i$ or choosing a parameter $R$ --- it simply uses the UCB strategy from (3a) for its sampling policy.
>
> **Other problem settings**: We agree that observing the diverse $\Delta_i$ case would be useful, and will include such an experiment in the final draft. We have also updated the final draft to emphasize the additional settings discussed in the appendix.
>
> **Moving materials into the main paper**: We did not do this only for reasons of space, but with the extra provided page, we will definitely take this suggestion seriously.

---

### Official Review · Reviewer_1bLS · 2021-07-15

**Rating:** 7
**Confidence:** 4

**Summary:**

The paper proposes a general framework for multiple hypothesis testing with data generated from bandit algorithms. In particular, the paper proposes a new e-value-based multiple testing method that 1) has a tighter FDR bound and 2) applies to more general settings, compared with previous p-valued-based methods (Jamieson and Jain, 2018). In a special case, the paper investigates the efficiency of its proposed algorithm and shows that the sample complexity needed for the TPR to be above the pre-specified threshold matches that of previous work (Jamieson and Jain, 2018). Additionally, the paper improves the FDR bound of the p-value-based method (Jamieson and Jain, 2018) by leveraging results from Su (2018).

**Limitations And Societal Impact:**

The authors have adequately addressed the limitations and potential negative societal impact of their work.

**Main Review:**

1. Conceptually, bringing the e-values (and the e-BH) to the bandit setting is quite a novel and natural idea, as one of the advantages of the e-value is that it deals with dependency well. On the technical side, it is not that surprising to see the FDR control established by leveraging the proof of e-BH.  I find the main technical difficulty here to be constructing (efficient) e-values---in general, it is not hard to construct valid e-values (since it is only required that its expectation is bounded by one) but it is non-trivial to obtain efficient e-values (it is possible that a valid e-value by construction can never be as large as 20).  In this paper, the authors provide an e-value constructing method based on Waudby-Smith and Ramdas (2021) that is proved to be efficient in a special case and appears to be efficient in numerical experiments.

2. In the paper, the authors describe their method as a meta-algorithm, while I tend to understand it as an inference method for data generated from bandit algorithms---it can be carried out post hoc since it does not interact with the data-collecting/arm-pulling process. Along this line, is it possible to inform decision-making with the e-values (in some special cases)? For example, in the case studied in Section 6, is it possible to design (based on e-values) a stopping time $\tau$, such that the output set $S_{\tau}$ achieves both FDR and TPR threshold (with $\tau$ being as small as possible). This should be similar to best arm identification, except that instead of finding the best arm, the aim is to quickly find a set of (mostly) non-trivial arms.

3. The exposition of the paper is in general clear, although I find some parts to be repetitive (e.g., p-BH requires independent structure).

4. Minor issues:
 - On page 2, in “...a strategy for for quickly...” there is an extra for.
 - On page 3, defining the canonical filtration, the notation (i,j,X_{i,j}) is a bit confusing as the reader might think of j as another arm. Maybe use s instead.
 - On page 4, “...but most important for is the...”, is there a typo?
 - Also page 4, what is an “admissible e-process”? Is it defined somewhere?
 - “This is *in* contrast with procedures involving p-variables”?
 - On page 5, the footnote mark 1 should be after the comma.
 - The paragraph under fact 2, it is not very clear what “these dependence structures” refer to.
 - On page 8, is there anything missing? why otherwise?



**Time Spent Reviewing:**

4 hours

---

> ### Author Response · Authors · 2021-08-10
> **Response to Reviewer 1bLS**
>
> **Constructing efficient e-values**: We agree that constructing efficient e-values plays a key role in making our framework effective. Recently, there have been many works proposing powerful e-processes for many different testing problems (e.g. [2],  Waudby-Smith and Ramdas 2021, amongst others).
> The GROW criterion in the first, or the Kelly criterion in the second, both suggest that the optimal e-variable E will maximize expected log(E) under the alternative hypothesis; this choice also minimizes the time till E crosses a threshold [1]. If the alternative is only loosely defined as a composite class (like Gaussians with nonzero means) then the second paper shows how to use ideas from betting to adaptively identify the right alternative and build up an e-process over time based on the data.
> Importantly, another recent work investigates the strengths of e-variables vs. p-variables [4] and seems to indicate each statistic can have more power in different scenarios. At the same time, e-variables are the only known way to test certain composite hypotheses [3] (i.e. the only known p-variables for testing these hypotheses are reciprocals of e-variables). Further, likelihood ratios are a well studied class of statistics that are also e-variables. Thus, using e-variables in our framework is a relatively robust choice as efficient e-variables can be constructed for a large variety of scenarios, and mathematical analysis in special cases is certainly rapidly evolving.
>
> **Can e-values guide decision making?** Yes, we propose an exploration strategy that selects an arm with the largest e-value that has yet to be rejected (Algorithm 3 in Appendix B). A future line of work is to understand how this type of exploration relates to existing methods such as UCB.
>
> **Can we construct a stopping time based on e-values, such that the output set achieves both an FDR and TPR threshold?**
> Just to clarify, no method in general can meet both FDR and TPR guarantees --- with or without e-values --- unless there is an additional minimum signal strength (or effect size) assumption; an arm that appears null after 1000 samples may well reveal itself to be a very very weak signal after 5000 samples. Thus, in general it is only possible to provide an FDR guarantee, not a FDR and TPR guarantee. For the latter, one needs some *separation* between the null and alternative hypotheses (the same thing as minimum signal strength or effect size); and in this case it is possible to derive stopping rules using e-values. This is implicit in our, and JJ's, sample complexity result, since the the complexity is dependent on $\Delta_i$, the gap in signal between non-null hypotheses and the null hypothesis.
> This is different from the best arm identification (BAI) setting (or other similar problems of identifying some property of arms that concerns its "ranking" relative to other arms), where the algorithm can terminate once it finds "enough" arms, because there is generally an upper bound on how many arms that need to be found to return a correct answer.
> Since we do not know the number of non-null arms in advance, or how many of our rejections are true discoveries, we can run our algorithm for longer to accrue more rejections, but we will never reach a similar "enough" criterion (unless we reject $k$ arms) where it is clear our algorithm must stop.
>
> **Minor issues**: Thank you for pointing out these errors and suggesting fixes. An admissible e-process refers to admissibility in the statistical sense i.e. there does not exist another e-process that is uniformly more powerful (faster growing) than an admissible e-process over all possible distributions - we have added this definition in the final draft. Under Fact 2, "these dependence structures" refer to bandit settings where we use facts about PRDS/independent p-variables vs. arbitrarily dependent p-variables --- we have updated the draft to clarify this point. On page 8, "otherwise" is used to contrast from the requirement of  $i \in \mathcal{H}_1$ in the previous sentence. In this situation, "otherwise" means "in the case that the $i$th hypothesis is null", and we have changed the sentence to explicitly state this.
>
>
> **References**
>
> [1] L. Breiman. Optimal Gambling Systems for Favorable Games. Berkeley Symposium on Mathematical Statistics and Probability, 1961.
>
> [2] P. Gr&uuml;nwald, R. de Heide, W. Koolen. Safe testing. 2019.
>
> [3] L. Wasserman, A. Ramdas, S. Balakrishnan. Universal inference. PNAS, 2020.
>
> [4] V. Vovk and R. Wang. E-values: Calibration, combination, and applications. Annals of Statistics (forthcoming), 2021.

---

### Official Review · Reviewer_VJPB · 2021-07-18

**Rating:** 6
**Confidence:** 4

**Summary:**

The paper presents an algorithmic framework for bandit multiple hypothesis setting that decouples exploration with summarization of evidence - where the summarization is done using either e-values or p-values. The main contribution is to show that the algorithm with e-values has some nice properties for controlling FDR, even when samples are dependent.


**Ethical Concerns:**

No ethical issues.

**Limitations And Societal Impact:**

Yes.

**Main Review:**

Novelty of contribution:
I am not convinced by the significance of the “unified framework” presented here. The authors present an algorithm with a specific notation that decouples the exploration criteria with evidence summarization. However, the results presented are still based on specific exploration criteria (UCB) and specific summarization method - p value or e-value based. I am not sure if the insights provided by the so-called unified framework are significant.

The main contribution seems to be the use of e-value instead of p-values in JJ, and Theorem 1 that demonstrates e-values can achieve cleaner control on FDR under dependence while giving similar performance in terms of TDR. However, there is some loss in sample complexity for achieving required TDR guarantees for the e-value based algorithm (Theorem 1 compared to Fact 4). It is not clear if that is an artifact of analysis technique or due to some fundamental shortcoming of using e-values compared to p-values.
Some discussion on this aspect would be useful.

Presentation

Overall, the paper is well written and easy to read. Since the framework is claimed to be an important contribution by the authors, one aspect of the problem setting that may be presented better is the performance criteria. For a large part of the paper (until Section 6), FDR seems to be the sole performance measure.  However, the guarantee $FDR \le \delta$ does not seem meaningful without a corresponding guarantee on how large TDR is. (One can always achieve small FDR by producing very small rejection sets). It will be good to introduce the objective of algorithm design in the beginning as : $FDR \le \delta$ and $TDR  \ge 1-\delta’$ with as small sample complexity as possible.

More concretely, when comparing procedures based on p-values and e-values in section 3, when the rejection sets produced are not identical for the two methods, ideally there should be some discussion on how large the corresponding rejection sets may be. Some guarantees are provided later in Section 6, but some conceptual discussion on TDR and sample complexity could be included in section 3 while discussing the impact on FDR.

Minor editorial comments:

On top of page 3, in definition of $H_1$ : ${\cal H}_1= [k]\backslash {\cal H}_0$

Theorem 1: “there will exist with T<..” Remove the extra word “with”

Summary:

Overall, the paper makes an interesting contribution to bandit multiple hypothesis testing using e-values instead of p-values. The guarantees achieved by the algorithm using e-values is better in terms of FDR but falls a bit short in terms of sample complexity to achieve required TDR. The significance of the unified framework is not clear.


**Time Spent Reviewing:**

4-5 hours

---

> ### Author Response · Authors · 2021-08-10
> **Response to Reviewer VJPB**
>
> **Novelty of framework**: The overarching goal of the framework is to allow for the exploration criteria to be freely chosen by the user based on the problem or priors about the setting, yet enable inference methods that are fully agnostic to the finer details of the problem setup; we agree that the evidence must be "restricted" to e-variables, or p-variables. We argue that this is not really a restriction and it is actually what brings utility to our framework! E-variables are a very new concept whose utility has only emerged in the past year or so, and hence the shift from p-variables to e-variables (and the corresponding benefit in theory and practice) for bandit multiple testing is not a trivial consequence; the development of the theory of e-variables is ongoing, and its implications for bandit problems has not been explored until now.
> Most importantly, the use of e-variables guarantees FDR control without any restrictions on the problem setup or choice of exploration method. Thus, we can (but need not) directly leverage existing best arm identification methods for exploration (Appendix B.1) or apply the framework to a streaming setting (Appendix B.2) and maintain provable FDR control. This separation of exploration and evidence is a result of a novel application of e-variables. This does not mean that smart exploration is not useful: it is, but there have been a tremendous number of works on that topic (that will continue to grow), and this work modularizes the two pieces so that any exploration component can be swapped in and out (in any problem setup) without affecting how the accumulation of evidence takes place. In fact, a significant amount of the work done by the JJ paper (several pages of analysis) was to prove FDR control in the very specific setting they considered; we show that a much simpler technique can be used for all settings, much beyond what JJ considered. This fact is very nontrivial, and immediately opens the door to new applications (and theoretical analysis in special cases), and our paper describes this general framework and its guarantees (both in general and the specific case of JJ).
>
> **Theorem 1 discussion**: Please see our root level comment --- we are able to eliminate the complexity gap between e-variables and p-variables.
>
>
> **Meaningfulness of performance measures**: We agree that guarantees concerning to power akin to TPR would enhance the paper. However, it is standard in the multiple testing literature to decouple the mathematical study of FDR control and power, because the latter requires many more assumptions on the fine details of the problem setup, but the former can often be done in a manner (like in our paper) that is agnostic to the inner details of the hypotheses and p-variables or e-variables. Our focus is not to discuss a specific bandit setting, but to provide a framework that is applicable to *all* bandit settings, regardless of dependence in the rewards, or number of arms observed/sampled on each step. In addition, our framework is applicable to any hypothesis testing problem, and not just the problem of identifying arms with positive means as discussed in Section 6. For instance, we could specify the hypothesis to be that the variance of the rewards is larger than 1, or any other hypothesis for which e-processes can be designed (a large and rapidly increasing class). Nor does our framework restrict the user to only focus on TPR e.g. the user could aim to ensure all non-null hypotheses are rejected with high probability instead. Further, we show that our framework can recover state-of-the-art sample complexity guarantees for TPR in a common setting. Thus, we open the way to exploring applications of our framework to different specific settings in future work.
>
>
> **Analysis of rejection sets**: We have added some discussion of the relationship between the rejections sets under p-variables vs. e-variables and provided some intuition for TPR and sample complexity, as suggested. We would like to emphasize that our framework concerns maintaining control of FDR at a predefined level at any stopping time. Thus, the sample complexity guarantee seen in Theorem 1 is dependent on the level of control $\delta$, and the tradeoff between FDR and TPR can be seen by varying this parameter. Consequently, lowering $\delta$ reduces FDR but results in larger sample complexity before a TPR of $1 - \delta$ is achieved. Note that we may also decouple $\delta$ for FDR and TPR from each other, but our theorems use the same $\delta$ in both for simplicity. As a result, the implicit goal is to optimize for sample complexity given a constraints on FDR and TPR (or maximize TPR while controlling FDR and having a fixed total sample budget), and this can only be done in restricted setups (like Section 6).
>
> **Minor editorial comments**: Thank you for identifying these issues - we have corrected them.

---

### Author Response · Authors · 2021-08-10
**Root level response**

Thanks to all the reviewers for their insightful comments and suggestions! Soon after submitting the main paper (but before the supplement deadline), we designed an e-process that has *matching sample complexity bounds with JJ (Fact 5)* for the independent, sub-Gaussian, and single arm case discussed in Section 6.  We had formally stated this in Theorem 2 in Appendix A.2.3 in the submitted supplement. Further, this e-process requires no additional parameters such as $R$, and can be directly used with the UCB exploration strategy formulated in (3a). Naturally, we have now moved this result to the main body of the paper and hope this addresses some of the concerns raised by reviewers with our algorithm. Also, we have corrected all the typos and minor issues that have been identified.

---

### Decision · Program_Chairs · 2021-09-27

**Decision:**

Accept (Poster)

**Comment:**

This is a borderline paper. As the reviewers point out there are some valuable contributions but also various shortcomings. For instance, it is not fully obvious what the benefit of the unified framework is, the objective of the algorithm design is not stated clearly, and the paper could be improved significantly in a revision with regard to the presentation. Despite these shortcomings I believe that the contributions are valuable and I recommend to accept the paper.